# Unified Projection-Free Algorithms for Adversarial DR-Submodular Optimization

**Mohammad Pedramfar**
School of Industrial Engineering
Purdue University
West Lafayette, IN 47907, USA
mpedramf@purdue.edu

**Yididiya Y. Nadew**
Department of Computer Science
Iowa State University
Ames, IA 50010, USA
yididiya@iastate.edu

**Christopher J. Quinn**
Department of Computer Science
Iowa State University
Ames, IA 50010, USA
cjquinn@iastate.edu

**Vaneet Aggarwal**
School of Industrial Engineering
Purdue University
West Lafayette, IN 47907, USA
vaneet@purdue.edu

## Abstract

This paper introduces unified projection-free Frank-Wolfe type algorithms for adversarial continuous DR-submodular optimization, spanning scenarios such as full information and (semi-)bandit feedback, monotone and non-monotone functions, different constraints, and types of stochastic queries. For every problem considered in the non-monotone setting, the proposed algorithms are either the first with proven sub-linear $\alpha$-regret bounds or have better $\alpha$-regret bounds than the state of the art, where $\alpha$ is a corresponding approximation bound in the offline setting. In the monotone setting, the proposed approach gives state-of-the-art sub-linear $\alpha$-regret bounds among projection-free algorithms in 7 of the 8 considered cases while matching the result of the remaining case. Additionally, this paper addresses semi-bandit and bandit feedback for adversarial DR-submodular optimization, advancing the understanding of this optimization area.

## 1 Introduction

The optimization of continuous adversarial DR-submodular functions has become increasingly prominent in recent years. This form of optimization represents an important subset of non-convex optimization problems at the forefront of machine learning and statistics. These challenges have numerous real-world applications like revenue maximization, mean-field inference, and recommendation systems, among others (Bian et al., 2019; Hassani et al., 2017; Mitra et al., 2021; Djolonga & Krause, 2014; Ito & Fujimaki, 2016; Gu et al., 2023; Li et al., 2023). The problems at hand can be conceptualized as a recurring game played between an optimizer and an adversary. In each round of this game, the optimizer makes an action selection, while the adversary selects a reward function. The optimizer is then allowed to query this reward function, either at any arbitrary point within the domain (full information) or specifically at the chosen action (in the case of semi-bandit/bandit scenarios). The adversary provides a noisy version of the gradient/value at the queried point. This framework gives rise to a set of significant challenges, varying based on the properties of the DR-submodular function, the constraint set, and the types of queries involved.

This paper presents a comprehensive investigation into online continuous adversarial DR-submodular optimization. There have been significant advances in recent years, though most research has predominantly focused on monotone (i.e., non-decreasing) objective functions and/or full information feedback via stochastic gradient queries. In contrast, our study encompasses a broader spectrum, addressing combinations of: (i) monotone or non-monotone functions, (ii) optimization under downward closed (or convex sets containing the origin) or general convex sets, (iii) gradient or value queries, and (iv) queries at arbitrary points or only the current action. See Table 1 for an

enumeration of the cases along with corresponding approximation ratios $\alpha$, query complexities (for full-information feedback), and $\alpha$-regret bounds for prior works and ours.

In this paper, we propose Frank-Wolfe based algorithms for this diverse family of problems. For many cases in Table 1, our algorithms are the first to achieve sub-linear regret or improve on the state of the art. For non-monotone objective functions (i.e., the bottom half of Table 1), our algorithms beat the state of the art for almost every combination of convex feasible regions (downward-closed or general), feedback models (full-information (with $T^\beta$ queries for various $\beta$), semi-bandit, bandit), and feedback type (exact/noisy gradient/value). See Fig. 1 for a visual depiction of the improved regret bound and query complexity trade offs for non-monotone functions with a downward closed feasible region and full-information feedback with (noisy) gradient queries. The single case for non-monotone objectives that our algorithms do not strictly improve on the prior works is for general convex sets with full-information feedback of $T^{1/2}$ gradients (per round), in which case the regret bound of our algorithm and that proposed by (Mualem & Feldman, 2023) both have $\tilde{O}(\sqrt{T})$ dependence. We note for $T^\beta$ queries with $0 \leq \beta < \frac{1}{2}$, our algorithm's regret bound is strictly better.

For monotone objective functions (i.e., the top half of Table 1), our algorithms achieve the first sublinear regret for general convex sets with full information value feedback and for general convex sets with bandit feedback. Our algorithms also achieve the first or better regret bounds than prior projection-free methods (those without "‡" symbols in Table 1) for all but one case, i.e. for general convex feasible regions with bandit feedback where we match the results of (Niazadeh et al., 2023).[1] See Appendix A for more discussion.

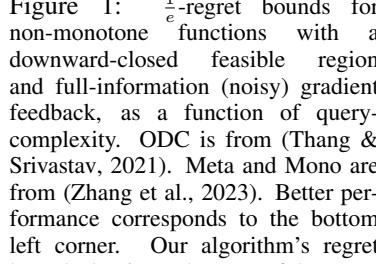

Figure 1: $\frac{1}{e}$-regret bounds for non-monotone functions with a downward-closed feasible region and full-information (noisy) gradient feedback, as a function of query-complexity. ODC is from (Thang & Srivastav, 2021). Meta and Mono are from (Zhang et al., 2023). Better performance corresponds to the bottom left corner. Our algorithm's regret bounds dominate the state of the art.

Our algorithms and most prior Frank-Wolfe based methods can rely on solving only linear optimization problems as subroutines (hence referred to as "projection-free"). Some of the prior works that use projected gradient ascent, in particular (Zhang et al., 2022; Chen et al., 2018b) (marked with a "‡" in Table 1), achieve superior regret bounds of $\tilde{O}(\sqrt{T})$ to other prior works and our algorithms. In some instances, solving a projection (or other non-linear optimization problems like (Wan et al., 2023; Thang & Srivastav, 2021)) as a sub-routine can be computationally expensive. Braun et al. (2022) identify matrix completion, routing and graph problems, and problems with matroid polytopes as example problems for which it is efficient to solve linear optimization problems but solving projections can be expensive. (Chen et al., 2018a) showed projected gradient ascent took between five to eight times longer than several Frank-Wolfe based methods even for small matrix completion tasks. Thus, for some large-scale problems, if the agent has limited per-round computational resources, the regret bounds achieved by projection-free methods (for which our methods match or improve on the state of the art) could represent the best "practically-achievable" regret-bounds.

The key contributions of this work can be summarized as follows:

1. We propose a unified framework for Frank-Wolfe type (projection-free) algorithms for adversarial continuous DR-submodular optimization, spanning scenarios with different types of feedback (full information, semi-bandit, and bandit; exact or stochastic), objective function classes (monotone and non-monotone), and convex feasible region geometries (downward-closed, origin feasible, general). In particular, we provide the first projection-free algorithm for online adversarial maximization of monotone functions over general convex sets for any feedback and oracle type.

2. For the class of non-monotone DR-submodular functions, our algorithms achieve the best (in some cases the first) sublinear $\alpha$-regret bounds for all feedback models and feasible region geometries considered.

---

[1] The result of (Niazadeh et al., 2023) is proven for monotone functions over $d$-dimensional downward-closed convex sets given a deterministic value oracle. However, as we discuss in Appendix C, replacing their shrunk constraint set with the construction presented in (Pedramfar et al., 2023) together with a more detailed analysis of their algorithms can be used to obtain the same regret bounds for monotone functions over all convex sets containing the origin when we only have access to a stochastic value oracle.

Table 1: Online DR-submodular optimization results.

| $F$ | Set | | Feedback | | Reference | Appx. | # of queries | $\log_T(\alpha\text{-regret})$ |
|---|---|---|---|---|---|---|---|---|
| Monotone | $0 \in \mathcal{K}$ | $\nabla F$ | Full Information | det. | (Chen et al., 2018b), (*) | $1-e^{-1}$ | $T^\beta(\beta \in [0,1/2])$ | $1-\beta$ |
| | | | | det. | (Niazadeh et al., 2023) | $1-e^{-1}$ | $T^\beta(\beta \in [0,1/2])$ | $1-\beta$ |
| | | | | stoch. | (Chen et al., 2018a), | $1-e^{-1}$ | $T^\beta(\beta \in [0,3/2])$ | $1-\beta/3$ |
| | | | | stoch. | (Zhang et al., 2019) | $1-e^{-1}$ | $1$ | $4/5$ |
| | | | | stoch. | (Liao et al., 2023) | $1-e^{-1}$ | $1$ | $3/4$ |
| | | | | stoch. | (Zhang et al., 2022) ‡ | $1-e^{-1}$ | $1$ | $1/2$ |
| | | | | stoch. | This paper | $1-e^{-1}$ | $T^\beta(\beta \in [0,1/2])$ | $2/3-\beta/3$ |
| | | | Semi-bandit | stoch. | This paper | $1-e^{-1}$ | - | $3/4$ |
| | | $F$ | Full Information | stoch. | This paper | $1-e^{-1}$ | $T^\beta(\beta \in [0,1/2])$ | $3/5-\beta/5$ |
| | | | Bandit | det. | (Zhang et al., 2019) | $1-e^{-1}$ | | $8/9$ |
| | | | | det. | (Niazadeh et al., 2023) | $1-e^{-1}$ | - | $5/6$ |
| | | | | det. | (Wan et al., 2023) ‡‡ | $1-e^{-1}$ | - | $3/4$ |
| | | | | stoch. | This paper | $1-e^{-1}$ | - | $5/6$ |
| | general | $\nabla F$ | Full Information | stoch. | This paper | $1/2$ | $T^\beta(\beta \in [0,1/2])$ | $2/3-\beta/3$ |
| | | | Semi-bandit | stoch. | (Chen et al., 2018b)‡ | $1/2$ | - | $1/2$ |
| | | | | | This paper | $1/2$ | - | $3/4$ |
| | | $F$ | Full Information | stoch. | This paper | $1/2$ | $T^\beta(\beta \in [0,1/2])$ | $3/5-\beta/5$ |
| | | | Bandit | stoch. | This paper | $1/2$ | - | $5/6$ |
| Non-Monotone | d.c. | $\nabla F$ | Full Information | stoch. | (Thang & Srivastav, 2021) | $e^{-1}$ | $T^\beta(\beta \in [0,3/4])$ | $1-\beta/3$ |
| | | | | | (Zhang et al., 2023) | $e^{-1}$ | $T^\beta(\beta \in [0,3/2])$ | $1-\beta/3$ |
| | | | | | (Zhang et al., 2023) | $e^{-1}$ | $1$ | $4/5$ |
| | | | | | This paper | $e^{-1}$ | $T^\beta(\beta \in [0,1/2])$ | $2/3-\beta/3$ |
| | | | Semi-bandit | stoch. | This paper | $e^{-1}$ | - | $3/4$ |
| | | $F$ | Full Information | stoch. | This paper | $e^{-1}$ | $T^\beta(\beta \in [0,1/2])$ | $3/5-\beta/5$ |
| | | | Bandit | det. | (Zhang et al., 2023) | $e^{-1}$ | - | $8/9$ |
| | | | | stoch. | This paper | $e^{-1}$ | - | $5/6$ |
| | general | $\nabla F$ | Full Information | stoch. | (Thang & Srivastav, 2021) | $(1-h)/3\sqrt{3}$ | $T^\beta(\beta > 0)$ | $1$ |
| | | | | | (Mualem & Feldman, 2023), (*) | $(1-h)/4$ | $T^\beta(\beta \in [0,1/2])$ | $1-\beta$ |
| | | | | | This paper | $(1-h)/4$ | $T^\beta(\beta \in [0,1/2])$ | $2/3-\beta/3$ |
| | | | Semi-bandit | stoch. | This paper | $(1-h)/4$ | - | $3/4$ |
| | | $F$ | Full Information | stoch. | This paper | $(1-h)/4$ | $T^\beta(\beta \in [0,1/2])$ | $3/5-\beta/5$ |
| | | | Bandit | stoch. | This paper | $(1-h)/4$ | - | $5/6$ |

Here $h := \min_{\mathbf{z} \in \mathcal{K}} \|\mathbf{z}\|_\infty$. The rows marked with (*) are special cases of our algorithms with appropriate hyperparameters. The rows marked with ‡ use gradient ascent, requiring potentially computationally expensive projections. ‡‡ (Wan et al., 2023) uses a convex optimization subroutine in each iteration. The logarithmic terms in regret are ignored. See Appendix A.2 for details.

3. For the class of monotone (i.e., non-decreasing) DR-submodular functions, our algorithms achieve the state of the art $\alpha$-regret bounds in 4 of the 8 cases.[2] Moreover, if we only compare with other projection-free algorithms, then we obtain the state of the art in 7 out of the 8 cases and match the result of (Niazadeh et al., 2023) in the last case.[1]

In addition to the enumerated list above, our technical novelties include (i) a novel combination of the idea of meta-actions and random permutations to obtain a new algorithm in full-information setting; (ii) handling stochastic (gradient/value) feedback *without* using variance reduction techniques like momentum, which in turn leads to state of the art regret bounds; and (iii) a unified approach that specializes to multiple scenarios considered in this paper. See Sections 3.1 and 3.2 and Appendix A.2 for more details. Table 1 describes the key comparisons of our works, where the related works are expanded on in Appendix A.

## 2 BACKGROUND AND NOTATION

We introduce some basic notions, concepts and assumptions which will be used throughout the paper. For any vector $\mathbf{x} \in \mathbb{R}^d$, $[\mathbf{x}]_i$ is the $i$-th entry of $\mathbf{x}$. We consider the partial order on $\mathbb{R}^d$ where $\mathbf{x} \le \mathbf{y}$ if and only if $[\mathbf{x}]_i \le [\mathbf{y}]_i$ for all $1 \le i \le d$. For two vectors $\mathbf{x}, \mathbf{y} \in \mathbb{R}^d$, the *join* of $\mathbf{x}$ and $\mathbf{y}$, denoted by $\mathbf{x} \vee \mathbf{y}$ and the *meet* of $\mathbf{x}$ and $\mathbf{y}$, denoted by $\mathbf{x} \wedge \mathbf{y}$, are defined

$$\mathbf{x} \vee \mathbf{y} := (\max\{[\mathbf{x}]_i, [\mathbf{y}]_i\})_{i=1}^d \quad \text{and} \quad \mathbf{x} \wedge \mathbf{y} := (\min\{[\mathbf{x}]_i, [\mathbf{y}]_i\})_{i=1}^d, \tag{1}$$

respectively. Clearly, we have $\mathbf{x} \wedge \mathbf{y} \le \mathbf{x} \le \mathbf{x} \vee \mathbf{y}$. We use $\mathbf{x} \odot \mathbf{y}$ for coordinate-wise multiplication. We use $\|\cdot\|$ to denote the Euclidean norm, and $\|\cdot\|_\infty$ to denote the supremum norm. In the paper, we consider a bounded convex domain $\mathcal{K}$ and w.l.o.g. assume that $\mathcal{K} \subseteq [0,1]^d$. We say that $\mathcal{K}$ is *downward-closed* (d.c.) if there is a point $\mathbf{u} \in \mathcal{K}$ such that for all $\mathbf{z} \in \mathcal{K}$, we have $\{\mathbf{x} \mid \mathbf{u} \le \mathbf{x} \le \mathbf{z}\} \subseteq \mathcal{K}$. Unless explicitly stated, we will assume that downward-closed convex sets contain the

---

[2]Algorithms for (semi-)bandit feedback can be used in full-information setting. Therefore (Chen et al., 2018b) obtains the state of the art in both semi-bandit and full-information setting for monotone functions over general convex set when given access to gradient oracles.

origin. The *diameter* $D$ of the convex domain $\mathcal{K}$ is defined as $D := \sup_{\mathbf{x},\mathbf{y}\in\mathcal{K}}\|\mathbf{x}-\mathbf{y}\|$. We use $\mathbb{B}_r(\mathbf{x})$ to denote the open ball of radius $r$ centered at $\mathbf{x}$. More generally, for a subset $X \subseteq \mathbb{R}^d$, we define $\mathbb{B}_r(X) := \bigcup_{\mathbf{x}\in X}\mathbb{B}_r(\mathbf{x})$. For an affine subspace $A$ of $\mathbb{R}^d$, we define $\mathbb{B}_r^A(X) := A \cap \mathbb{B}_r(X)$. We will use $\mathbb{R}_+^d$ to denote the set $\{\mathbf{x} \in \mathbb{R}^d | \mathbf{x} \geq 0\}$. For any set $X \subseteq \mathbb{R}^d$, the affine hull of $X$, denoted by $\mathrm{aff}(X)$, is defined to be the intersection of all affine subsets of $\mathbb{R}^d$ that contain $X$. The *relative interior* of a set $X$ is defined by

$$\mathrm{relint}(X) := \{\mathbf{x} \in X \mid \exists\varepsilon > 0, \mathbb{B}_\varepsilon^{\mathrm{aff}(X)}(\mathbf{x}) \subseteq X\}.$$

It is well known that for any non-empty convex set $\mathcal{K}$, the set $\mathrm{relint}(\mathcal{K})$ is always non-empty. We will always assume that the feasible set contains at least two points and therefore the dimension $d' := \dim(\mathcal{K}) = \dim(\mathrm{aff}(\mathcal{K})) \geq 1$, otherwise the optimization problem is trivial.

A set function $f : \{0,1\}^d \to \mathbb{R}$ is called *submodular* if for all $\mathbf{x},\mathbf{y} \in \{0,1\}^d$ with $\mathbf{x} \geq \mathbf{y}$,

$$f(\mathbf{x} \vee \mathbf{a}) - f(\mathbf{x}) \leq f(\mathbf{y} \vee \mathbf{a}) - f(\mathbf{y}), \qquad \forall \mathbf{a} \in \{0,1\}^d. \tag{2}$$

Submodular functions can be generalized over continuous domains. A function $F : [0,1]^d \to \mathbb{R}$ is called *DR-submodular* if for all vectors $\mathbf{x},\mathbf{y} \in [0,1]^d$ with $\mathbf{x} \leq \mathbf{y}$, any basis vector $\mathbf{e}_i = (0,\cdots,0,1,0,\cdots,0)$ and any constant $c > 0$ such that $\mathbf{x} + c\mathbf{e}_i \in [0,1]^d$ and $\mathbf{y} + c\mathbf{e}_i \in [0,1]^d$,

$$F(\mathbf{x} + c\mathbf{e}_i) - F(\mathbf{x}) \geq F(\mathbf{y} + c\mathbf{e}_i) - F(\mathbf{y}). \tag{3}$$

Note that if a function $F$ is differentiable then the diminishing-return (DR) property (3) is equivalent to $\nabla F(\mathbf{x}) \geq \nabla F(\mathbf{y})$ for $\mathbf{x} \leq \mathbf{y}$ with $\mathbf{x},\mathbf{y} \in [0,1]^d$. A function $F : \mathcal{D} \to \mathbb{R}$ is $M_1$-*Lipschitz continuous* if for all $\mathbf{x},\mathbf{y} \in \mathcal{D}$, $\|F(\mathbf{x}) - F(\mathbf{y})\| \leq M_1\|\mathbf{x}-\mathbf{y}\|$. A differentiable function $F : \mathcal{D} \to \mathbb{R}$ is $M_2$-*smooth* if for all $\mathbf{x},\mathbf{y} \in \mathcal{D}$, $\|\nabla F(\mathbf{x}) - \nabla F(\mathbf{y})\| \leq M_2\|\mathbf{x} - \mathbf{y}\|$. A *DR-submodular* $F$ is monotone if $F(\mathbf{x}) \geq F(\mathbf{y})$ for all $\mathbf{x} \geq \mathbf{y}$.

## 2.1 PROBLEM SETUP

Adversarial bandit optimization problems can be formalized as a repeated game between an optimizer and an adversary. The game lasts for $T$ rounds and $T$ is known to both players. In $t$-th round, the optimizer chooses an action $\mathbf{x}_t$ from an action set $\mathcal{K}$, then the adversary chooses a reward function $F_t \in \mathcal{F}$. We assume that the function class $\mathcal{F}$ has functions that map $\mathcal{K}$ to a bounded interval $[0, M_0] \subseteq \mathbb{R}$. We consider the setting with *oblivious adversary* where the choice of the sequence of functions $F_t$ is not affected by the choice of the optimizer. In other words, we may assume that the adversary chooses the sequence $\{F_t\}_{t=1}^T$ before the first action of the optimizer.

Before we discuss different forms of feedback, we first formally define the notion of oracle. A stochastic *non-oblivious* value oracle for the function $F : \mathcal{K} \to \mathbb{R}$ is a tuple $(\mathfrak{Z}_0, p_0, \tilde{F})$ where $\mathfrak{Z}_0$ is an arbitrary measure space, $p_0 : \mathfrak{Z}_0 \times \mathcal{K} \to \mathbb{R}$ is a non-negative measurable function such that $\int_{\mathfrak{Z}_0} p_0(\mathbf{z};\mathbf{x})d\mathbf{z} = 1$ for each $\mathbf{x} \in \mathcal{K}$ and $\tilde{F} : \mathfrak{Z}_0 \times \mathcal{K} \to \mathbb{R}$ is a measurable function such that

$$F(\mathbf{x}) = \mathbb{E}_{\mathbf{z}\sim p_0(\cdot;\mathbf{x})}[\tilde{F}(\mathbf{z}, \mathbf{x})],$$

for all $\mathbf{x} \in \mathcal{K}$. We will use $\tilde{F}(\mathbf{x})$ to denote the random variable $\tilde{F}(\mathbf{x}, \mathbf{z})$ where $\mathbf{z}$ is a random variable samples according to the distribution $p_0(\cdot;\mathbf{x})$. Such an oracle would be called an *oblivious* oracle when $p_0(\cdot;\mathbf{x})$ is independent of the choice of $\mathbf{x}$. We consider the more general setting where we only have access to a non-oblivious oracle, i.e., where $p_0(\cdot;\mathbf{x})$ may depend on $\mathbf{x}$.

Similarly, a non-oblivious gradient oracle for the function $F : \mathcal{K} \to \mathbb{R}$ is a tuple $(\mathfrak{Z}_1, p_1, \tilde{\nabla}F)$ where $\mathfrak{Z}_1$ is an arbitrary measure space, $p_1 : \mathfrak{Z}_1 \times \mathcal{K} \to \mathbb{R}$ is a non-negative measurable function such that $\int_{\mathfrak{Z}_1} p_1(\mathbf{z};\mathbf{x})d\mathbf{z} = 1$ for each $\mathbf{x} \in \mathcal{K}$ and $\tilde{\nabla}F : \mathfrak{Z}_1 \times \mathcal{K} \to \mathbb{R}$ is a measurable function such that

$$\nabla F(\mathbf{x}) = \mathbb{E}_{\mathbf{z}\sim p_1(\cdot;\mathbf{x})}[\tilde{\nabla}F(\mathbf{z}, \mathbf{x})],$$

for all $\mathbf{x} \in \mathcal{K}$. Similarly, we will use $\tilde{\nabla}F(\mathbf{x})$ to denote the random variable $\tilde{\nabla}F(\mathbf{x}, \mathbf{z})$ where $\mathbf{z}$ is a random variable sampled according to the distribution $p_1(\cdot;\mathbf{x})$.

**Assumption 1.** *We assume that the functions $F_t : [0,1]^d \to \mathbb{R}$ are DR-submodular, first-order differentiable, non-negative, bounded by $M_0$, $M_1$-Lipschitz, and $M_2$-smooth for some values of $M_0, M_1, M_2 < \infty$. Note that this implies that $\|\nabla F_t(\mathbf{x})\| \leq M_1$. Moreover, we also assume that we either have access to a value oracle bounded by $B_0$ or a gradient oracle bounded by $B_1$ for some values of for some $B_0, B_1 < \infty$.*

**Remark 1.** *The proposed algorithm does not need to know the values of $M_0$, $M_1$, $M_2$, $B_0$ or $B_1$, in advance. However these variables appear in the regret bounds. Note that we always have $B_0 \geq M_0$ and $B_1 \geq M_1$. Exact oracles are special cases of stochastic oracles with $B_0 = M_0$ and $B_1 = M_1$. When we have access to exact oracles, the performance of the proposed algorithms does not change beyond the replacement of $B_0$ with $M_0$ and $B_1$ with $M_1$.*

We consider different forms of feedback to the optimizer:

1. **Full Information with gradient query oracle**: In this feedback model, the optimizer is allowed to query a stochastic non-oblivious gradient oracle $\tilde{\nabla} F_t(\mathbf{x})$ for $F_t : \mathcal{K} \to \mathbb{R}$ at multiple points. We assume that the optimizer can query $F_t$ a total of $T^\beta$ times, where $\beta \geq 0$ gives a range from constant queries to infinite queries.

2. **Full Information with value query oracle**: Same as the previous case, but the adversary reveals a stochastic non-oblivious value oracle $\tilde{F}_t(\mathbf{x})$.

3. **Semi-Bandit**: In this feedback model, the adversary reveals a gradient sample $\tilde{\nabla} F_t(\mathbf{y}_t)$ for the specific action $\mathbf{y}_t$ taken, where $\tilde{\nabla} F_t$ is a stochastic non-oblivious gradient oracle for $F_t$.

4. **Noisy Bandit:** In this feedback model, the optimizer can only observe a sample of $\tilde{F}_t(\mathbf{y}_t)$, where $\tilde{F}_t$ is a stochastic non-oblivious value oracle for $F_t$. Such feedback model is a generalization of what is called *full-bandit feedback* in the literature. In the full-bandit feedback setting, the optimizer observes the exact value of $F_t(\mathbf{y}_t)$.

We note that the full information feedback can query at any point in $\mathcal{K}$, while the semi-bandit and bandit feedback can only query at $\mathbf{y}_t$ in time $t$. Also note that the distributions $p_0$ and $p_1$, described in the definition of stochastic oracles, may depend on the function $F_t$. We will use a superscript, i.e., $p_0^t$ and $p_1^t$, to specify the function in question.

Please note that even when dealing with offline scenarios, it is NP-hard to solve a DR-submodular maximization problem (Bian et al., 2017b). For the problems we consider, however, there are polynomial time approximation algorithms. We let $\alpha$ denote corresponding approximation ratios. Thus, the goal of this work is to minimize the $\alpha$-regret, which is defined as:

$$\mathcal{R}_\alpha = \alpha \max_{\mathbf{y} \in \mathcal{K}} \sum_{t=1}^{T} F_t(\mathbf{y}) - \sum_{t=1}^{T} F_t(\mathbf{y}_t) \tag{4}$$

In Appendix A.1, we discuss the best known approximation ratios in different settings.

**Remark 2.** *Any algorithm designed for semi-bandit setting may be trivially applied in full-information setting with a gradient oracle. Similarly, any algorithm designed for bandit setting may be applied in full-information setting with a value oracle.*

**Remark 3.** *As a special case, if all of the functions $F_t$ are equal, then the semi-bandit setting we consider reduces to online stochastic continuous DR-submodular maximization. See Table 2 in Appendix for the list of previous results in this setting for (i) monotone/non-monotone function, (ii) constraint set choices, or (iii) bandit/semi-bandit feedback. These results achieve the same regret guarantees as in (Pedramfar et al., 2023), and thus match the state of art for projection-free algorithms in all cases.*

## 3 PROPOSED ALGORITHMS

In this section, we describe the proposed algorithm with different forms of feedback and the different problem setups (based on the properties of the functions and the feasible set). For efficient description of the algorithm, we first divide the problem setup into four categories:

(A) The functions $\{F_t\}_{t=1}^{T}$ are monotone DR-submodular and $\mathbf{0} \in \mathcal{K}$.

(B) The functions $\{F_t\}_{t=1}^{T}$ are non-monotone DR-submodular and $\mathcal{K}$ is a downward closed set containing 0.

(C) The functions $\{F_t\}_{t=1}^{T}$ are monotone DR-submodular and $\mathcal{K}$ is a general convex set.

(D) The functions $\{F_t\}_{t=1}^{T}$ are non-monotone DR-submodular and $\mathcal{K}$ is a general convex set.

The four divisions here for the problem provide different approximation ratios $\alpha$ for the problem. Thus, the algorithm steps and the proofs change with these cases. For combining definitions in the different forms of feedback, we define

$$\text{oracle-adv}(\mathbf{d}, \mathbf{x}) := \begin{cases} \mathbf{d} \odot (1 - \mathbf{x}) & \text{(B)}; \\ \mathbf{d} & \text{otherwise}, \end{cases}, \text{update}(\mathbf{x}, \mathbf{v}, \underline{\mathbf{u}}) = \begin{cases} \mathbf{x} + \frac{1}{K}(\mathbf{v} - \underline{\mathbf{u}}) & \text{(A)}; \\ \mathbf{x} + \frac{1}{K}(\mathbf{v} - \underline{\mathbf{u}}) \odot (1 - \mathbf{x}) & \text{(B)}; \\ (1 - \varepsilon_C)\mathbf{x} + \varepsilon_C \mathbf{v} & \text{(C)}; \\ (1 - \varepsilon_D)\mathbf{x} + \varepsilon_D \mathbf{v} & \text{(D)}, \end{cases}$$

where $\varepsilon_C = \frac{\log(K)}{2K}$ and $\varepsilon_D = \frac{\log(2)}{K}$. We also define the approximation ratio $\alpha$ as $1 - \frac{1}{e}$ for case (A), $\frac{1}{e}$ for case (B), $\frac{1}{2}$ for case (C), and $\frac{1-h}{4}$ for case (D), where $h := \min_{\mathbf{z} \in \mathcal{K}} \|\mathbf{z}\|_\infty$. Further,

$$\text{grad-estimate}(F, \mathbf{x}, \mathbf{u}) := \begin{cases} \tilde{\nabla} F(\mathbf{x}) & \text{gradient oracle}; \\ \frac{d'}{\delta} \tilde{F}(\mathbf{x} + \delta \mathbf{u})\mathbf{u} & \text{value oracle}. \end{cases}$$

In the following subsections, we divide the proposed algorithm into different feedback scenarios.

### 3.1 FULL INFORMATION

There are four main ideas used in Algorithm 1 that allows us to obtain the desired regret bounds.

**1. Offline bounds** *Given a submodular function $F$, sequence of vectors $(\mathbf{v}^{(k)})_{k=1}^K$ in the convex set $\mathcal{K}$ and a sequence of points $\mathbf{x}^{(k+1)} = \text{update}(\mathbf{x}^{(k)}, \mathbf{v}^{(k)})$ for $k \in [K]$, for any $\mathbf{x}^* \in \mathcal{K}$, we may bound $\alpha F(\mathbf{x}^*) - F(\mathbf{x}^{(K+1)})$ from above by the sum of a known term and a linear combination of $\langle \text{oracle-adv}(\nabla F(\mathbf{x}^{(k)}), \mathbf{x}^{(k)}), \mathbf{v}^{(k)} - \mathbf{x}^* \rangle$, for $k \in [K]$.*

See Lemma 8 for the exact statement for different cases. This lemma captures the core idea behind all Frank-Wolfe type algorithms for DR-submodular maximization. In particular, if we choose

$$\mathbf{v}^{(k)} \in \text{argmax}_{\mathbf{v} \in \mathcal{K}} \langle \text{oracle-adv}(\nabla F(\mathbf{x}^{(k)}), \mathbf{x}^{(k)}), \mathbf{v} \rangle,$$

we recover the results in offline setting with access to a deterministic gradient oracle. Lemma 8 is a reformulation of some of the ideas presented in (Bian et al., 2017b;a; Zhang et al., 2023; Du, 2022; Mualem & Feldman, 2023) and (Pedramfar et al., 2023).

**2. Meta actions** Having $L = 1$, $\delta = 0$, and access to gradient oracles corresponds to the idea of meta-actions (without using Ideas 3 and 4). The idea of meta-actions, proposed in (Streeter & Golovin, 2008) for discrete submodular functions, was first used for continuous DR-submodular maximization in (Chen et al., 2018b). This idea allows us to convert offline algorithm into online algorithms. To be precise, let us consider the first iteration and the first objective function $F_1$ of our online optimization setting. Note that $F_1$ remains unknown until the algorithm commits to a choice. If we were in the offline setting, we could have chosen

$$\mathbf{v}^{(k)} \in \text{argmax}_{\mathbf{v} \in \mathcal{K}} \langle \text{oracle-adv}(\nabla F_1(\mathbf{x}^{(k)}), \mathbf{x}^{(k)}), \mathbf{v} \rangle,$$

to obtain the desired regret bounds. The idea of meta-actions is to mimic this process in an online setting as follows. We run $K$ instances of an online linear optimization (OLO) algorithm, $\{\mathcal{E}^{(k)}\}_{k=1}^K$.

---

**Input :** smoothing radius $\delta$, shrunk constraint set $\hat{\mathcal{K}}$, horizon $T$, block size $L$, number of linear maximization oracles $K$, online linear maximization oracles on $\hat{\mathcal{K}}$: $\mathcal{E}^{(1)}, \cdots, \mathcal{E}^{(K)}$, number of blocks $Q = T/L$.

**for** $q = 1, 2, \ldots, Q$ **do**
  Pick any $\underline{\mathbf{u}} \in \text{argmin}_{\mathbf{x} \in \hat{\mathcal{K}}} \|\mathbf{x}\|_\infty$
  $\mathbf{x}_q^{(1)} \leftarrow \underline{\mathbf{u}}$
  **for** $k = 1, 2, \ldots, K$ **do**
    Let $\mathbf{v}_q^{(k)} \in \hat{\mathcal{K}}$ be the output of $\mathcal{E}^{(k)}$ in round $q$.
    $\mathbf{x}_q^{(k+1)} \leftarrow \text{update}(\mathbf{x}_q^{(k)}, \mathbf{v}_q^{(k)}, \underline{\mathbf{u}})$
  **end**
  $\mathbf{x}_q \leftarrow \mathbf{x}_q^{(K+1)}$
  Let $(t_{q,1}, \ldots, t_{q,L})$ be a random permutation of $\{(q-1)L + 1, \ldots, qL\}$
  **for** $t = (q-1)L + 1, \ldots, qL$ **do**
    Play $\mathbf{y}_t = \mathbf{x}_q$ and obtain the reward $F_t(\mathbf{y}_t)$
    Find the corresponding $l \in [L]$ such that $t = t_{q,l}$
    **for** $k \in [K]$ *such that* $k \equiv l \pmod{L}$ **do**
      If we have a value oracle, sample $\mathbf{u}_q^{(k)} \sim \mathbb{S}^{d-1} \cap \mathcal{L}_0$ uniformly, otherwise $\mathbf{u}_q^{(k)} \leftarrow \mathbf{0}$
      $\mathbf{d}_q^{(k)} \leftarrow \text{grad-estimate}(F_{t_{q,l}}, \mathbf{x}_q^{(k)}, \mathbf{u}_q^{(k)})$
      $\mathbf{g}_q^{(k)} \leftarrow \text{oracle-adv}(\mathbf{d}_q^{(k)}, \mathbf{x}_q^{(k)})$
      Pass $\mathbf{g}_q^{(k)}$ as the adversarially chosen vector to $\mathcal{E}^{(k)}$
    **end**
  **end**
**end**

**Algorithm 1:** Generalized Meta-Frank-Wolfe

---

Here the number $K$ denotes the number of iterations of the offline Frank-Wolfe algorithm that we intend to mimic. Thus, to maximize $\langle \text{oracle-adv}(\nabla F_1(\mathbf{x}^{(k)}), \mathbf{x}^{(k)}), \cdot \rangle$, we simply use $\mathcal{E}^{(k)}$. Once the function $F_1$ is revealed to the algorithm, it knows each linear maximization oracle "adversaries" $\{\text{oracle-adv}(\nabla F_1(\mathbf{x}^{(k)}), \mathbf{x}^{(k)})\}_{k=1}^{K}$. Now, we simply feed each online algorithm $\mathcal{E}^{(k)}$ with the reward $\{\langle \text{oracle-adv}(\nabla F_1(\mathbf{x}^{(k)}), \mathbf{x}^{(k)}), \cdot \rangle\}_{k=1}^{K}$. We repeat this process for each subsequent function $\{F_t\}_{t \geq 2}$. This idea, combined with Idea 1, allows us to obtain the desired $\alpha$-regret bounds.

**Remark 4.** *We assume that every instance $\mathcal{E}^{(k)}$ has the following behavior and guarantee. In every block $1 \leq q \leq Q$, the oracle $\mathcal{E}^{(k)}$ selects a vector $\mathbf{v}_q^{(k)}$ and then the adversary reveals a vector $\mathbf{g}_q^{(k)}$ to the oracle that was chosen independently of $\mathbf{v}_q^{(k)}$. The OLO oracle guarantees that $\sum_{q=1}^{Q} \langle \mathbf{g}_q^{(k)}, \mathbf{x}^* - \mathbf{v}_q^{(k)} \rangle \leq \mathcal{R}_Q^{\mathcal{E}^{(k)}}$, for some regret function $\mathcal{R}_Q^{\mathcal{E}^{(k)}}$. One possible choice for such an oracle is Follow-the-Perturbed-Leader by (Kalai & Vempala, 2005) that guarantees $\mathcal{R}_Q^{\mathcal{E}^{(k)}} \leq CDB\sqrt{Q}$ where $D$ is the diameter of $\mathcal{K}$, $B = \max_{q,k} \|\mathbf{g}_q^{(k)}\|$ and $C > 0$ is a constant. It follows from the definition of* grad-estimate *that if we have access to gradient oracles, then $B \leq B_1$, while if we have access to value oracles, then $B \leq \frac{d'}{\delta} B_0$.*

**3. Random permutations** Using random permutations allows us to use less queries at the cost of increased regret. In the context of DR-submodular maximization, this idea was first used in Mono-Frank-Wolfe algorithm in (Zhang et al., 2019). The Mono-Frank-Wolfe corresponds to Algorithm 1 when $K = L$ and we have access to a gradient oracle. Here we describe this idea in the general setting where we allow $K \neq L$, while we still assume access to a gradient oracle. We start by dividing the $T$ functions into $Q = T/L$ blocks of length $L$. We define $\bar{F}_q$ as the average of functions in the $q$-th block. For each block $q$, we pick a random permutation $(t_{q,1}, \ldots, t_{q,L})$ of $\{(q-1)L+1, \ldots, qL\}$ uniformly from the set of all of its permutations. The key insight is that for all $(q-1)L < t \leq qL$, the expected value of $F_t$ is $\bar{F}_q$. Therefor we can estimate $\nabla \bar{F}_q$ using information obtained from functions $F_t$ for $(q-1)L < t \leq qL$ which allows us to apply the idea of meta-actions on the sequence of functions $\{\bar{F}_q\}_{q=1}^{Q}$.

**4. Smoothing trick** When we do not have access to a gradient oracle, we rely on samples from a value oracle to estimate the gradient. The "smoothing trick" (Flaxman et al., 2005; Hazan et al., 2016; Agarwal et al., 2010; Shamir, 2017; Zhang et al.,

---

**Input :** smoothing radius $\delta$, shrunk constraint set $\hat{\mathcal{K}}$, horizon $T$, block size $L$, the number of exploration steps per block $K \leq L$, online linear maximization oracles on $\hat{\mathcal{K}}$: $\mathcal{E}^{(1)}, \cdots, \mathcal{E}^{(K)}$, number of blocks $Q = T/L$.

**for** $q = 1, 2, \ldots, Q$ **do**
  Pick any $\underline{\mathbf{u}} \in \text{argmin}_{\mathbf{x} \in \hat{\mathcal{K}}} \|\mathbf{x}\|_\infty$
  $\mathbf{x}_q^{(1)} \leftarrow \underline{\mathbf{u}}$
  **for** $k = 1, 2, \ldots, K$ **do**
    Let $\mathbf{v}_q^{(k)} \in \hat{\mathcal{K}}$ be the output of $\mathcal{E}^{(k)}$ in round $q$
    $\mathbf{x}_q^{(k+1)} \leftarrow \text{update}(\mathbf{x}_q^{(k)}, \mathbf{v}_q^{(k)}, \underline{\mathbf{u}})$
  **end**
  $\mathbf{x}_q \leftarrow \mathbf{x}_q^{(K+1)}$
  Let $(t_{q,1}, \ldots, t_{q,L})$ be a random permutation of $\{(q-1)L+1, \ldots, qL\}$
  **for** $t = (q-1)L+1, \ldots, qL$ **do**
    **if** $t \in \{t_{q,1}, \cdots, t_{q,K}\}$ **then**
      Find the corresponding $k \in [K]$ such that $t = t_{q,k}$
      If we have a value oracle, sample
        $\mathbf{u}_q^{(k)} \sim \mathbb{S}^{d-1} \cap \mathcal{L}_0$ uniformly, otherwise $\mathbf{u}_q^{(k)} \leftarrow \mathbf{0}$
      Play $\mathbf{y}_t = \mathbf{y}_{t_{q,k}} = \mathbf{x}_q^{(k)} + \delta \mathbf{u}_q^{(k)}$ for $F_t$ (i.e., $F_{t_{q,k}}$)
      // Exploration
      $\mathbf{d}_q^{(k)} \leftarrow \text{grad-estimate}(F_{t_{q,k}}, \mathbf{x}_q^{(k)}, \mathbf{u}_q^{(k)})$
      $\mathbf{g}_q^{(k)} \leftarrow \text{oracle-adv}(\mathbf{d}_q^{(k)}, \mathbf{x}_q^{(k)})$
      Pass $\mathbf{g}_q^{(k)}$ as the adversarially chosen vector to $\mathcal{E}^{(k)}$
    **else**
      Play $\mathbf{y}_t = \mathbf{x}_q$ for $F_t$              // Exploitation
    **end**
  **end**
**end**

**Algorithm 2:** Generalized (Semi-)Bandit-Frank-Wolfe

---

2019; Chen et al., 2020; Zhang et al., 2023; Niazadeh et al., 2023; Pedramfar et al., 2023) involves averaging through spherical sampling around a given point. Here we use a variant that was introduced in (Pedramfar et al., 2023).

**Definition 1** (Smoothing Trick). *For a function $F : \mathcal{D} \to \mathbb{R}$ defined on $\mathcal{D} \subseteq \mathbb{R}^d$, its $\delta$-smoothed version $\hat{F}$ is given as*

$$\hat{F}_\delta(\mathbf{x}) := \mathbb{E}_{\mathbf{z} \sim \mathbb{B}_\delta^{\text{aff}(\mathcal{D})}(\mathbf{x})}[F(\mathbf{z})] = \mathbb{E}_{\mathbf{v} \sim \mathbb{B}_1^{\text{aff}(\mathcal{D})-\mathbf{x}}(\mathbf{0})}[F(\mathbf{x} + \delta \mathbf{v})], \quad (5)$$

*where $\mathbf{v}$ is chosen uniformly at random from the $\dim(\mathrm{aff}(\mathcal{D}))$-dimensional ball $\mathbb{B}_1^{\mathrm{aff}(\mathcal{D})-\mathbf{x}}(\mathbf{0})$. Thus, the function value $\hat{F}_\delta(\mathbf{x})$ is obtained by "averaging" $F$ over a sliced ball of radius $\delta$ around $\mathbf{x}$.*

The power of the smoothing trick lies in the facts that it is a good approximation of the original function (Lemma 1) and there is a simple one-point gradient estimator for the smoothed version of the function (Lemma 2). We will drop the subscript $\delta$ when there is no ambiguity.

Note that the domain of $\hat{F}$ is smaller than the domain of $F$. Therefore, our ability to estimate the gradient is limited to a smaller region compared to the entire feasible set. This limitation should be considered when developing an algorithm for DR-submodular maximization. The notion of "shrunk constraint set" (Zhang et al., 2019; 2023; Niazadeh et al., 2023; Pedramfar et al., 2023) was designed to address this concern. Here we use the variant designed by (Pedramfar et al., 2023). Formally, we choose a point $\mathbf{c} \in \mathrm{relint}(\mathcal{K})$ and a real number $r > 0$ such that $\mathbb{B}_r^{\mathrm{aff}(\mathcal{K})}(\mathbf{c}) \subseteq \mathcal{K}$. Then, for a given $\delta < r$, we define $\hat{\mathcal{K}}_\delta^{\mathbf{c},r} := (1 - \frac{\delta}{r})\mathcal{K} + \frac{\delta}{r}\mathbf{c}$. Clearly if $\mathcal{K}$ is downward-closed, then so is $\hat{\mathcal{K}}_\delta^{\mathbf{c},r}$. We will use the notation $\hat{\mathcal{K}}$ to denote this set when there is no ambiguity.

Putting these ideas together, in order to maximize $F$ over $\mathcal{K}$ with a value oracle, we restrict ourselves to the shrunk constraint set $\hat{\mathcal{K}}$ and maximize $\hat{F}$ over this set using the one-point gradient estimator.

### 3.2 (SEMI-)BANDIT FEEDBACK

In the (semi-)bandit feedback setting, we only observe the value or gradient at the point where the action is taken. To adapt Algorithm 1 to this setting, we start by assuming $K \leq L$. In each block, there are $L$ functions that are used to update $K$ linear maximization oracles. Therefore, we may choose $K$ exploration steps in each block where we take actions according to what we queried in Algorithm 1. These actions are informative and allow us to carry out similar analysis to the full information setting. In the remaining $L - K$ steps, we exploit our knowledge of the best action so far to minimize total $\alpha$-regret. See Algorithm 2 for pseudo-code.

## 4 $\alpha$-REGRET GUARANTEES

For brevity, we define $\mathcal{R}^u$ as

$$\mathcal{R}^u = \left(M_0 + (3 + D)M_1\right)\frac{T\delta}{r} + \begin{cases} (8M_0 + M_2D^2\log(K)^2)\frac{T}{8K} + LCDB\sqrt{Q}\log(K) & (C); \\ (M_0 + 2M_2D^2)\frac{T}{4K} + LCDB\sqrt{Q} & \text{Otherwise,} \end{cases}$$

where $B \leq B_1$ if we have access to a gradient oracle and $B \leq \frac{d'}{\delta}B_0$ otherwise and $C$ is the constant in Remark 4.

**Theorem 1.** *Using Algorithm 1, we have $\mathbb{E}[\mathcal{R}_\alpha] \leq \mathcal{R}^u$.*

The proof is in Appendix F. Note that the number of times any function $F_t$ is queried in Algorithm 1 is $K/L$. Let $\beta$ be a real number such that $T^\beta = K/L$. Given a gradient oracle, for any choice of $0 \leq \beta \leq \frac{1}{2}$, we may set $\delta = 0$, $L = T^{\frac{1-2\beta}{3}}$ and therefore $K = T^{\frac{1+\beta}{3}}$ and $Q = T/L = T^{\frac{2+2\beta}{3}}$, to obtain $\mathbb{E}[\mathcal{R}_\alpha] = O(T^{\frac{2-\beta}{3}}\log(T)^2)$ in case (C), and $O(T^{\frac{2-\beta}{3}})$, otherwise. The special case $L = 1$ corresponds to Meta-Frank-Wolfe (Chen et al., 2018b;a; Zhang et al., 2023; Thang & Srivastav, 2021; Mualem & Feldman, 2023) while the special case $L = K$ corresponds to Mono-Frank-Wolfe (Zhang et al., 2019; 2023). Similarly, given a value oracle, for any choice of $0 \leq \beta \leq \frac{1}{2}$, by setting $\delta = T^{-\frac{2+\beta}{5}}$, $L = T^{\frac{2-4\beta}{5}}$ and therefore $K = T^{\frac{2+\beta}{5}}$ and $Q = T/L = T^{\frac{3+4\beta}{3}}$, we see that $\mathbb{E}[\mathcal{R}_\alpha] = O(T^{\frac{3-\beta}{5}}\log(T)^2)$ in Case (C), and $O(T^{\frac{3-\beta}{5}})$, otherwise.

**Theorem 2.** *Using Algorithm 2, we have $\mathbb{E}[\mathcal{R}_\alpha] \leq \mathcal{R}^u + 2M_0QK$.*

The proof is in Appendix G. In particular, given a gradient oracle, we may set $\delta = 0$, $K = T^{1/4}$ and $L = T^{1/2}$ and therefore $Q = T^{1/2}$, to obtain $\mathbb{E}[\mathcal{R}_\alpha] = O(T^{3/4}\log(T)^2)$ in Case (C) and $O(T^{3/4})$, otherwise. Similarly, when given a value oracle, if we set $K = T^{1/6}$, $L = T^{1/3}$, $\delta = T^{-1/6}$ and therefore $Q = T^{2/3}$, we see that $\mathbb{E}[\mathcal{R}_\alpha] = O(T^{5/6}\log(T)^2)$ in Case (C) and $O(T^{5/6})$, otherwise.

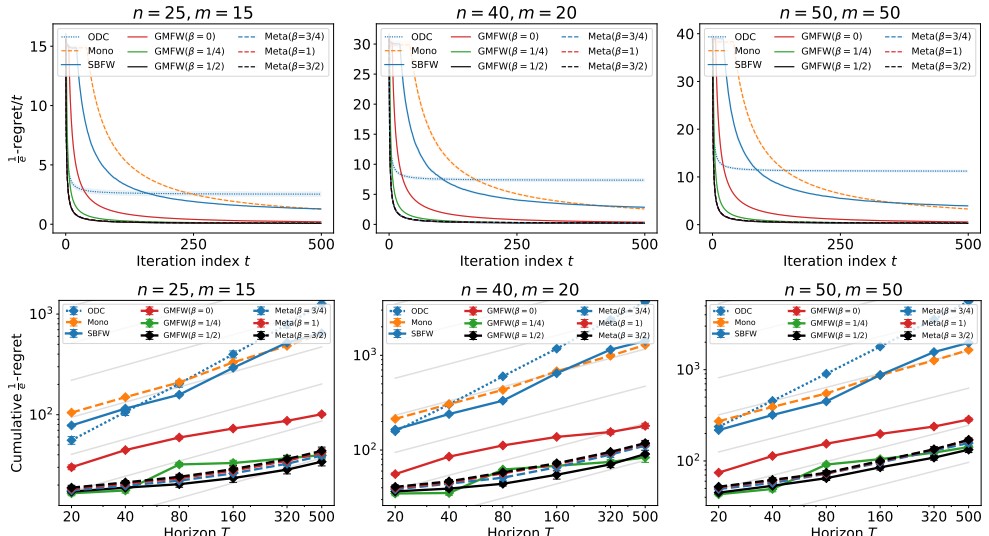

Figure 2: Empirical regret plots for the experiments. The top row depicts time-averaged regret for each round $t$ for a horizon of $T = 500$. The bottom row depicts cumulative regret for multiple horizons with logarithmic scaling. Grey lines in the bottom row represent $y = aT^{1/2}$ curves for different $a$ for visual reference. Colors correspond to regret bounds (i.e. black for $\tilde{O}(T^{1/2})$). Our methods (solid lines) use significantly fewer queries and less computation than baselines (dashed and dotted lines) with similar regret bounds and achieve better regret than baselines using similar numbers of queries and computation.

## 5 EXPERIMENTS

We test our online continuous DR-submodular maximization algorithms for non-monotone objectives, a downward-closed feasible region, and both full-information and semi-bandit gradient feedback. We briefly describe the setup and highlight key results. See Appendix H for more details. We use online non-convex/non-concave non-monotone quadratic maximization following (Bian et al., 2017a; Chen et al., 2018b; Zhang et al., 2023), randomly generating linear inequalities to form a downward closed feasible region and for each round $t$ we generate a quadratic function $F_t(\mathbf{x}) = \frac{1}{2}\mathbf{x}^\top \mathbf{H}\mathbf{x} + \mathbf{h}^\top \mathbf{x} + c$. Similar to (Zhang et al., 2023), we considered three pairs $(n, m)$ of dimensions $n$ and number of constraints $m$, $\{(25, 15), (40, 20), (50, 50)\}$.

We ran three online algorithms from prior works, **ODC** from (Thang & Srivastav, 2021), **Mono** (full information single query) from (Zhang et al., 2023), and **Meta**($\beta$) from (Zhang et al., 2023), where we used query parameters $\beta = \{3/4, 1, 3/2\}$; here and in the following we only explicitly mention the query parameter so that there are $T^\beta$ queries per round and other algorithm parameters implicit. We ran our Algorithm 1 (**GMFW**($\beta$) for short) with query parameter $\beta = \{0, 1/4, 1/2\}$ and our semi-bandit Algorithm 2 (**SBFW** for short). Fig. 1 depicts regret bound and query complexity trade offs for full-information methods.

Figure 2 shows both averaged regret within runs for a fixed horizon (top row) and cumulative regret for different horizons, averaged over 10 independent runs. See Fig. 2's caption for a description. Average run-times for a horizon of $T = 100$ are displayed in Table 3 in Appendix H. Major differences in run-times is in large part due to the number of online linear maximization oracles used, which is in part related to the number of per-round queries.

In each experiment, our **GMFW**($\beta = 1/2$) (black solid line) performs the best overall, despite using significantly fewer gradient queries and significantly less computation than any of the **Meta** algorithms. Our **GMFW**($\beta = 0$) (red solid line) performs better than the baseline **Mono** (orange dashed line; designed for the same amount of feedback).

ACKNOWLEDGMENTS

This work was supported in part by the National Science Foundation under grants CCF-2149588 and CCF-2149617. We also thank the authors of (Zhang et al., 2023) for sharing their code.

REPRODUCIBILITY STATEMENT

Source code for our algorithms is available at `https://github.com/yididiyan/unified-dr-submodular`. All assumptions are included in the main paper in Assumption 1. All proofs are included in the appendices.

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

# A DETAILS OF RELATED WORKS

## A.1 OFFLINE DR-SUBMODULAR MAXIMIZATION

The problem of continuous DR-submodular maximization was considered by Bian et al. (2017b) where they introduced a variant of the Frank-Wolfe algorithm for maximizing a monotone DR-submodular function a convex set containing the origin. They assumed access to a deterministic gradient oracle and guaranteed an optimal $(1 - \frac{1}{e})$-approximation with an additive error of $O(\varepsilon)$ using $O(1/\varepsilon)$ oracle queries. We will refer to this guarantee as having the query complexity of $O(1/\varepsilon)$. Feige (1998) proved that *discrete* submodular maximization for monotone functions with cardinality constraint is NP-hard for any approximation coefficient higher than $(1 - 1/e)$. Using the results of (Feige, 1998), Bian et al. (2017b) showed that, for continuous DR-submodular functions over a downward-closed set, obtaining an approximation coefficient higher than $(1 - 1/e)$ is NP-hard. Karbasi et al. (2019) proposed a Frank-Wolfe variant with access to a stochastic gradient oracle with *known distribution* that obtains $(1 - 1/e)$-approximation coefficient with $O(1/\varepsilon^2)$ oracle queries. They also proved that, in this setting, the oracle complexity of $O(1/\varepsilon^2)$ is optimal.

Later, another Frank-Wolfe variant with different update rules was proposed by Bian et al. (2017a) for non-monotone DR-submodular maximization over downward-closed convex sets given a deterministic gradient oracle, obtaining a $\frac{1}{e}$-approximation with the same query complexity of $O(1/\varepsilon)$. Gharan & Vondrák (2011) showed that obtaining an approximation ratio of higher than $0.491$ requires exponentially many queries of the oracle. Very recently, Chen et al. (2023) obtained a $0.385$-approximation with a query complexity of $O(1/\varepsilon)$. It is not known if that coefficient is optimal; finding an algorithm with the optimal approximation coefficient in this setting remains an open problem.

The first projection based algorithm for DR-submodular maximization was proposed by (Hassani et al., 2017) for monotone functions over a general convex set. They obtained a $\frac{1}{2}$-approximation guarantee with a query complexity of $O(1/\varepsilon)$ for deterministic gradient oracles and $O(1/\varepsilon^2)$ for stochastic gradient oracles. They provided an example showing that projected gradient ascent can not obtain a better approximation coefficient in this setting. Currently, it remains an open problem whether it is possible to achieve a higher approximation coefficient with sub-exponential query complexity. Later Pedramfar et al. (2023) used a Frank-Wolfe variant to obtain $\frac{1}{2}$-approximation guarantees for maximizing a monotone function over a general convex set with $\tilde{O}(1/\varepsilon)$ query complexity for deterministic gradient oracles.

Hassani et al. (2017) also constructed an example showing that directly replacing a deterministic gradient oracle with a stochastic one in (Bian et al., 2017b) can result in arbitrarily bad results. To solve this issue, Mokhtari et al. (2020) used a variance reduction technique, namely momentum, which reduces variance but introduces bias in a controlled manner, to obtain a query complexity of $O(1/\varepsilon^3)$ for monotone functions over convex sets containing the origin. Similarly, they also obtained a query complexity of $O(1/\varepsilon^3)$ for non-monotone functions over downward-closed convex sets. Zhang et al. (2022) proposed a novel gradient-based approach for monotone functions over a convex set containing the origin to obtain an approximation coefficient of $(1 - 1/e)$ with $O(1/\varepsilon^2)$ query complexity.

Vondrák (2013) proved that, for non-monotone functions over a general convex set, no constant approximation ratio can be guaranteed in sub-exponential time. However, as demonstrated in (Dürr et al., 2019), we may obtain approximation ratios that depend on the geometry of the convex set. Specifically, given a deterministic gradient oracle for a non-monotone function over a general convex set, they proposed an algorithm that obtains a $\frac{1}{3\sqrt{3}}(1 - h)$-approximation of the optimal value with an oracle complexity of $O(e^{\sqrt{d/\varepsilon}})$ where $h := \min_{\mathbf{z} \in \mathcal{K}} \|\mathbf{z}\|_\infty$ and $d$ is the dimension of the domain of the function. Later, Du et al. (2022) obtained a $\frac{(1-h)}{4}$-approximation guarantee with the same oracle complexity and Du (2022) obtained a $\frac{(1-h)}{4}$-approximation guarantee with $O(1/\varepsilon)$ query complexity. Mualem & Feldman (2023) proved that the approximation coefficient $\frac{(1-h)}{4}$ is optimal in this setting and obtaining any higher approximation coefficient requires exponentially many oracle queries.

The problem of DR-submodular maximization when given access to a value oracle was first considered by Chen et al. (2020) where they obtained a $(1 - 1/e)$-approximation coefficient with a query complexity of $O(1/\varepsilon^3)$ (for deterministic value oracles) and $O(1/\varepsilon^5)$ (for stochastic value oracles) for monotone functions over general convex sets. However, as mentioned in Appendix B of (Pedramfar et al., 2023), they require access to the value oracle outside the feasible set. Pedramfar et al. (2023) used a comprehensive approach to obtained $\tilde{O}(1/\varepsilon)$ for deterministic gradient oracle, $\tilde{O}(1/\varepsilon^3)$ for stochastic gradient oracle and deterministic value oracle and $\tilde{O}(1/\varepsilon^5)$ for stochastic value oracles with an approximation coefficient of $(1 - 1/e)$ for monotone functions over convex sets containing the origin, $1/e$ for non-monotone functions over downward closed sets, $1/2$ for monotone functions over general convex sets and $\frac{(1-h)}{4}$ for non-monotone functions over general convex sets.

Note that the problem of maximizing a non-monotone DR-submodular function over a box, i.e. $[0,1]^d$, can be solved using a discretization of the domain and applying algorithms designed for discrete settings. For example, Bian et al. (2019); Niazadeh et al. (2020) used discretization to obtain $\frac{1}{2}$-approximations of the optimal value. We have not considered this setting in our work since such discretization techniques have not been successfully applied in more general settings where the convex set is not a box.

## A.2 ONLINE DR-SUBMODULAR MAXIMIZATION

Chen et al. (2018b) considered the problem of online adversarial continuous DR-submodular maximization for monotone functions over a convex set that contains the origin. They obtained $(1 - 1/e)$ approximation guarantees given full information deterministic feedback. More specifically, they extended the idea of *meta-actions*, introduced by (Streeter & Golovin, 2008) for the discrete setting, to continuous setting and introduced the first version of Meta-Frank-Wolfe for continuous DR-submodular maximization. They obtained $(1 - 1/e)$-regret of $O(T^{1-\beta})$ when allowed to query each function $T^\beta$ times for $0 \le \beta \le 1/2$. It was shown by Hassani et al. (2017) that a direct usage of unbiased estimates of the gradients in Frank-Wolfe type algorithms can lead to arbitrarily bad solutions in the context of offline submodular maximization. This led Chen et al. (2018b) to consider a different approach when only unbiased estimates of gradients are available. In particular, they showed that online gradient ascent can obtain $\frac{1}{2}$-regret of $O(T^{1/2})$ in the semi-bandit feedback setting, where the algorithm is allowed to query the stochastic gradient oracle at the point where the action is taken. Later, Chen et al. (2018a) extended the Meta-Frank-Wolfe algorithm to work with stochastic gradient oracle using the momentum technique, described in (Mokhtari et al., 2020) for offline setting, to control the variance of the stochastic oracle. For monotone functions over convex sets containing the origin, they obtained $(1 - 1/e)$-regret of $O(T^{1-\beta/3})$ when allowed to query each function $T^\beta$ times for $0 \le \beta \le 3/2$. Zhang et al. (2019) continued this line of work by combining meta-actions, momentum and a novel idea of random permutations (see Idea 3.1) to obtain $(1 - 1/e)$-regret of $O(T^{4/5})$ when allowed to query the stochastic gradient oracle once for each function. Their algorithm, called Mono-Frank-Wolfe, requires access to the oracle at some point other than the point where the action is chosen. Using similar ideas, they also introduced Bandit-Frank-Wolfe which obtains $(1 - 1/e)$-regret of $O(T^{8/9})$, when a deterministic value oracle is available, for monotone functions over convex sets containing the origin.

For non-monotone function maximization over downward-closed convex sets, Thang & Srivastav (2021) introduced a novel technique which uses an online non-convex optimization oracles referred to an online vee-learning oracle (as opposed to online linear maximization oracles used in most other Frank-Wolfe based methods) to obtain a $\frac{1}{e}$-regret bound of $O(T^{1-\beta/3})$ using $T^\beta$ oracle queries per function for all $0 \le \beta \le 3/4$. Note that while their online vee-learning oracle is solving a non-convex optimization problem, it can obtain its guarantees in a projection-free manner. Later Zhang et al. (2023) extended the results of (Zhang et al., 2019) to the setting of non-monotone functions over downward-closed convex sets. More specifically, they showed that using their update rules, Meta-Frank-Wolfe obtains $\frac{1}{e}$-regret of $O(T^{1-\beta/3})$ when allowed to query each function $T^\beta$ times for $0 \le \beta \le 3/2$, Mono-Frank-Wolfe obtains $\frac{1}{e}$-regret of $O(T^{4/5})$ with only a single access to a stochastic gradient oracle and Bandit-Frank-Wolfe obtains $O(T^{8/9})$ with access to a deterministic value oracle. For monotone functions over convex sets containing the origin, Zhang et al. (2022) developed a boosting framework which allowed them to obtain $(1 - 1/e)$-regret of $O(T^{1/2})$ with

only a single access to a stochastic gradient oracle. Note that their result is not in the semi-bandit feedback setting, since they require access to the gradient oracle at points other than the points where the actions are taken. Also note that their algorithm uses projected gradient ascent which involves the computationally expensive projection operation. (Liao et al., 2023) adapted the boosting framework to obtain a projection-free Frank-Wolfe type algorithm with $(1-1/e)$-regret of $O(T^{3/4})$ with only a single access to a stochastic gradient oracle. (Wan et al., 2023) also adapted the boosting framework to obtain an algorithm for deterministic bandit feedback setting with $(1 - 1/e)$-regret of $O(T^{3/4})$. However, while their algorithm does not directly use projections, they require solving a convex optimization subroutine at each iteration which could be computationally expensive. (Niazadeh et al., 2023) showed that for monotone functions over convex sets that contain the origin, their algorithm obtains $(1 - 1/e)$-regret of $O(T^{5/6})$ in the deterministic bandit feedback setting. Recall that (Hassani et al., 2017) showed that a direct usage of unbiased estimates of the gradients in Frank-Wolfe type algorithms can lead to arbitrarily bad solutions in the context of offline submodular maximization. However, in the algorithm proposed by (Niazadeh et al., 2023), the exact gradient is directly replaced with a one-point estimate of the gradient without using momentum or similar variance reduction techniques.

For non-monotone functions over a general convex set, (Thang & Srivastav, 2021) considered a variant of Meta-Frank-Wolfe with different update rules to obtain a $\frac{1}{3\sqrt{3}}(1 - h)$-regret of $O(T/\log(T))$ using $T^\beta$ number of queries per function for any constant $\beta > 0$. While their algorithm used momentum, (Mualem & Feldman, 2023) considered the same algorithm without using momentum and used an improved analysis to obtain $\frac{(1-h)}{4}$-regret of $O(T^{1-\beta})$ when allowed to query each function $T^\beta$ times for $0 \leq \beta \leq 1/2$. While their result seems to match the first Meta-Frank-Wolfe result of (Chen et al., 2018b), it should be noted that this result works for stochastic gradient oracles, while the result of (Chen et al., 2018b) was for deterministic gradient oracles.

As previously mentioned, a notable challenge in this problem domain has been transitioning from a deterministic gradient oracle to a stochastic one. The existing solutions range from using the momentum technique of (Mokhtari et al., 2020), for example in (Chen et al., 2018a; Zhang et al., 2019; 2022), to using projection-based methods. One of our main technical novelties is our refined analysis which allows us to demonstrate that we can move from a deterministic gradient oracle to a stochastic one while keeping the same order of regret bounds. It also allowed us to move from deterministic value oracles to stochastic value oracles, both in the full-information setting and in the (semi-)bandit setting. This is the first work that considers online adversarial DR-submodular maximization with noisy (semi-)bandit feedback. As a special case, if we assume the sequence of functions $F_t$ is constant, then this problem reduces to online stochastic DR-submodular maximization with (semi-)bandit feedback. This problem has been studied for monotone functions over convex sets containing the origin in (Chen et al., 2018a) where they obtained $\frac{1}{e}$-regret of $O(T^{2/3})$. For monotone functions over general convex sets in semi-bandit setting, (Hassani et al., 2017) obtained a $\frac{1}{2}$-regret of $O(T^{1/2})$ using projected gradient ascent. Finally, (Pedramfar et al., 2023) obtained results in all cases as described in Table 2. Our unified online algorithm, when applied to the non-adversarial setting, matches the state of the art among all projection-free algorithms. If we include the projection based algorithms in our comparison, then we match the state of the art in 7 out of 8 cases.

## B  COMPARISON FOR ONLINE STOCHASTIC DR-SUBMODULAR MAXIMIZATION

Table 2: Online stochastic DR-submodular optimization.

| $F$ | Set | Feedback | Reference | Coef. $\alpha$ | $\alpha$-Regret |
|---|---|---|---|---|---|
| Monotone | $0 \in \mathcal{K}$ | $\nabla F$ | (Chen et al., 2018a)†, | $1/e$ | $O(T^{2/3})$ |
| | | | (Pedramfar et al., 2023) | $1 - 1/e$ | $O(T^{3/4})$ |
| | | | This paper | $1 - 1/e$ | $O(T^{3/4})$ |
| | | $F$ | (Pedramfar et al., 2023) | $1 - 1/e$ | $O(T^{5/6})$ |
| | | | This paper | $1 - 1/e$ | $O(T^{5/6})$ |
| | general | $\nabla F$ | (Hassani et al., 2017) ‡ | $1/2$ | $O(T^{1/2})$ |
| | | | (Pedramfar et al., 2023) | $1/2$ | $\tilde{O}(T^{3/4})$ |
| | | | This paper | $1/2$ | $\tilde{O}(T^{3/4})$ |
| | | $F$ | (Pedramfar et al., 2023) | $1/2$ | $\tilde{O}(T^{5/6})$ |
| | | | This paper | $1/2$ | $\tilde{O}(T^{5/6})$ |
| Non-mono. | d.c. | $\nabla F$ | (Pedramfar et al., 2023) | $1/e$ | $O(T^{3/4})$ |
| | | | This paper | $1/e$ | $O(T^{3/4})$ |
| | | $F$ | (Pedramfar et al., 2023) | $1/e$ | $O(T^{5/6})$ |
| | | | This paper | $1/e$ | $O(T^{5/6})$ |
| | general | $\nabla F$ | (Pedramfar et al., 2023) | $\frac{1-h}{4}$ | $O(T^{3/4})$ |
| | | | This paper | $\frac{1-h}{4}$ | $O(T^{3/4})$ |
| | | $F$ | (Pedramfar et al., 2023) | $\frac{1-h}{4}$ | $O(T^{5/6})$ |
| | | | This paper | $\frac{1-h}{4}$ | $O(T^{5/6})$ |

This table compares the different results for the expected $\alpha$-regret for online stochastic DR-submodular maximization for the under bandit and semi-bandit feedback. In this setting, our algorithm matches the state of the art in 7 of the 8 cases. When only compared with other projection-free algorithms, our result matches the state of the art in all cases. † the analysis in (Chen et al., 2018a) has an error (Appendix B in (Pedramfar et al., 2023)). ‡ (Hassani et al., 2017) uses gradient ascent, requiring potentially computationally expensive projections.

## C  COMMENTS ON PREVIOUS RESULTS IN LITERATURE

**Constraint set:**  Following (Pedramfar et al., 2023), one of our assumptions is that we are only allowed to query the oracles within the constraint set. This assumption allows for a more detailed exploration of the problem space and clarifies the reason why some algorithms for monotone DR-submodular maximization obtain $\frac{1}{2}$ approximation coefficients while others obtain $(1 - 1/e)$. In Appendix A, while discussing some of the related works, we have cited results that encompass a broader scope than what was initially claimed in the original paper, while some have seemingly narrower scope. In this section, we delve into how the proofs or algorithms from those works may be adapted to the more general setting we have considered.

As mentioned in Appendix A in (Pedramfar et al., 2023), the result of (Bian et al., 2017b) for monotone functions are presented for downward-closed sets, but the same proof can be used in the setting where the convex set only contains the origin and is not necessarily downward closed. The regret bounds of Bandit-Frank-Wolfe in (Zhang et al., 2019) are presented for downward-closed convex sets that are $d$-dimensional. By replacing their construction of shrunk constraint set with the construction described in (Pedramfar et al., 2023), one can show that their result applies to all convex sets that contain the origin. The Bandit-Frank-Wolfe of Zhang et al. (2022) is presented for non-monotone functions over $d$-dimensional downward closed sets. Similarly, by improving their construction of shrunk constraint set, one can show that their results apply to all downward-closed sets. The result of (Niazadeh et al., 2023) for monotone functions is also presented for $d$-dimensional downward closed sets which can be similarly modified to apply to all convex sets that contain the origin. The results of (Wan et al., 2023) are for $d$-dimensional convex sets that contain the origin

and not lower dimensional convex sets. We believe a similar approach may be applied to extend their result to all convex sets containing the origin.

For a convex set $\mathcal{K} \subseteq [0, 1]^d$, let $\mathcal{K}^*$ denote the convex hull of $\mathcal{K} \cup \{\mathbf{0}\}$. If $F$ is a monotone DR-submodular function over $[0, 1]^d$, then the maximum of $F$ over the feasible set $\mathcal{K}$ is the same as the maximum of $F$ over the feasible set $\mathcal{K}^*$. Therefore, these problems are identical if we have no further assumptions. However, if we assume that we may only query $F$ within the feasible set, then these problems become different. In fact, as discussed in (Pedramfar et al., 2023), the best known approximation coefficient for the first problem is $\frac{1}{2}$ while the optimal approximation coefficient for the second problem is $(1 - 1/e)$. The results of (Mokhtari et al., 2020) for monotone functions are presented for general convex sets, but if we assume that we can only query the oracle within the feasible set, then their algorithm and proofs apply when the convex set contains the origin. The same is true for the Meta-Frank-Wolfe algorithm of (Chen et al., 2018b) and (Chen et al., 2018a) and the One-Shot-Frank-Wolfe algorithm of (Chen et al., 2018a). Note that the One-Shot-Frank-Wolfe is claimed to obtain $(1 - 1/e)$-approximation, but as mentioned in Appendix B of (Pedramfar et al., 2023), the proof only results in an approximation coefficient of $1/e$.

Similarly, Zhang et al. (2022) considered the general convex set. However, their boosting technique involves a line integral over the line segment connecting the origin to the points in the feasible set and the algorithm queries the oracle on that line segment. Hence we classify their result as an algorithm that maximizes monotone DR-submodular functions over convex sets containing the origin.

**Bandit feedback:** As we have mentioned in Appendix A, this is the first work that considers the online adversarial DR-submodular maximization with noisy bandit feedback. We have mentioned four papers in Table 1 that consider a similar problem, but with deterministic feedback, namely (Zhang et al., 2019; 2023; Niazadeh et al., 2023; Wan et al., 2023). We believe a similar analysis to ours may be used to extend their results to the noisy feedback setting, likely without much change in their algorithms.

**Boundedness of stochastic gradient/value oracle:** Another one of our assumptions is that the stochastic gradient/value oracles are bounded. At first glance, this seems more restrictive than some of the previous works, such as Chen et al. (2018a), Thang & Srivastav (2021) and Zhang et al. (2023) which only claim to require that the stochastic gradient oracle should have bounded variance. However, a more careful review of their assumptions and proofs reveals that they require extra assumptions for their proofs to work. In particular, if they assume the boundedness of the stochastic gradient oracle, then their results hold up to a constant multiplicative factor.

In Chen et al. (2018a), Assumption 3 only requires that the stochastic gradient oracle should have bounded variance. However, in the last paragraph of their Section 4.1, they claim that "From Theorem 1, we observe that by setting $K = T$ and choosing a projection-free online linear optimization oracle with $\mathcal{R}(T) = O(\sqrt{T})$, such as Follow the Perturbed Leader (Cohen & Hazan, 2015), both regrets are bounded above by $O(\sqrt{T})$." The cited paper does not contain such an algorithm that can be applied to this setting. Moreover, to the best of our knowledge, there is no online linear optimization algorithm that can obtain a regret bound of $\mathcal{R}(T) = O(\sqrt{T})$ without any extra assumptions. On the other hand, as stated in Remark 4, there are algorithms that obtain $\mathcal{R}(T) = O(DB\sqrt{T})$ where $D$ is the diameter of the convex set and $B$ is an upper bound on the norm of the vectors that are passed to the algorithm. Therefore, if we add the assumption that the stochastic gradient oracle is bounded by some constant $B$, then the regret bound $O(T^{1-\beta/3})$ holds.

Similarly, for Meta-Frank-Wolfe and Mono-Frank-Wolfe algorithms of Zhang et al. (2023), Assumption 2 only requires that the stochastic gradient oracle should have bounded variance and Assumption 1-(iii) assumes access to an online linear maximization oracle with the regret bound of $O(\sqrt{T})$. As in Chen et al. (2018a), if we add the assumption that the stochastic gradient oracle is bounded by $B$, then their regret bounds, i.e. Theorems 1 and 2 in their paper, hold up to a multiplicative factor of $DB$.

In Thang & Srivastav (2021), in the case of general convex sets, the same issue appears in Theorem 3 and could be resolved similarly by adding the assumption of the boundedness of the stochastic gradient oracle.

Finally, in Thang & Srivastav (2021), in the case of downward-closed convex sets, Theorem 1 only assumes we have access to a stochastic gradient oracle with bounded variance. However, in the portion of proof of Theorem 1 before Claim 1, we can see that $\mathbf{d}^t_{\ell'}$ corresponds to $\mathbf{a}^t$ in Algorithm 1 and Lemma 4. (Note within the proof of Claim 1, $\nabla F^t(\mathbf{x}^t_\ell)$ corresponds to $\mathbf{a}_\ell$ in Lemma 3) Hence, the assumption $\|\mathbf{a}^t\| \le G$ stated in Lemma 4 does not follow from the stated assumptions in Theorem 1, e.g. the assumption that the norm of the gradients $\|\nabla F^t\|$ are bounded by $G$. Instead, it requires $\mathbf{d}^t_{\ell'}$ to be bounded which is satisfied if the stochastic gradient oracle is bounded.

## D  USEFUL LEMMAS

Here we state some lemmas from the literature that we will need in our analysis of DR-submodular functions.

**Lemma 1** (Lemma 11 of (Pedramfar et al., 2023)). *If $F : \mathcal{D} \to \mathbb{R}$ is DR-submodular, bounded by $M_0$, $M_1$-Lipschitz continuous, and $M_2$-smooth, then so is its smoothed version $\hat{F}$, and for any $\mathbf{x} \in \mathcal{D}$ such that $\mathbb{B}^{\mathrm{aff}(\mathcal{D})}_\delta(\mathbf{x}) \subseteq \mathcal{D}$, we have*

$$\|\hat{F}(\mathbf{x}) - F(\mathbf{x})\| \le \delta M_1.$$

*Moreover, if $F$ is monotone, then so is $\hat{F}_\delta$.*

**Lemma 2** (Lemma 13 in (Pedramfar et al., 2023)). *Let $\mathcal{D} \subseteq \mathbb{R}^d$ and $A := \mathrm{aff}(\mathcal{D})$. Also let $A_0$ be the translation of $A$ that contains $0$. Assume $F : \mathcal{D} \to \mathbb{R}$ is a Lipschitz continuous function and let $\hat{F}$ be its $\delta$-smoothed version. For any $\mathbf{z} \in \mathcal{D}$ such that $\mathbb{B}^A_\delta(\mathbf{z}) \subseteq \mathcal{D}$, we have*

$$\mathbb{E}_{\mathbf{u} \sim S^{d-1} \cap A_0} \left[ \frac{d'}{\delta} F(\mathbf{z} + \delta \mathbf{u}) \mathbf{u} \right] = \nabla \hat{F}(\mathbf{z}),$$

*where $d' = \dim(A_0)$.*

**Lemma 3** (Lemma 14 in (Pedramfar et al., 2023)). *Let $\mathcal{K} \subseteq [0,1]^d$ be a convex set containing the origin. Then for any choice of $\mathbf{c}$ and $r$ with $\mathbb{B}^{\mathrm{aff}(\mathcal{K})}_r(\mathbf{c}) \subseteq \mathcal{K}$, we have*

$$\arg\min_{\mathbf{z} \in \hat{\mathcal{K}}} \|\mathbf{z}\|_\infty = \frac{\delta}{r}\mathbf{c} \quad and \quad \min_{\mathbf{z} \in \hat{\mathcal{K}}} \|\mathbf{z}\|_\infty \le \frac{\delta}{r}.$$

**Lemma 4** (Lemma 15 in (Pedramfar et al., 2023)). *Let $\mathcal{K}$ be an arbitrary convex set, $D := \mathrm{diam}(\mathcal{K})$ and $\delta' := \frac{\delta D}{r}$. We have*

$$\mathbb{B}^{\mathrm{aff}(\mathcal{K})}_\delta(\hat{\mathcal{K}}) \subseteq \mathcal{K} \subseteq \mathbb{B}^{\mathrm{aff}(\mathcal{K})}_{\delta'}(\hat{\mathcal{K}}).$$

**Lemma 5** (Lemma 2.2 of (Mualem & Feldman, 2023)). *For any two vectors $\mathbf{x}, \mathbf{y} \in [0,1]^d$ and any continuously differentiable non-negative DR-submodular function $F$ we have*

$$F(\mathbf{x} \vee \mathbf{y}) \ge (1 - \|\mathbf{x}\|_\infty)F(\mathbf{y}).$$

**Lemma 6** (Lemma 6 of (Zhang et al., 2023)). *For any two vectors $\mathbf{x}, \mathbf{y} \in [0,1]^d$ and any continuously differentiable non-negative DR-submodular function $F$ we have*

$$F(\mathbf{y} + (\mathbf{1} - \mathbf{y}) \odot \mathbf{x}) \ge (1 - \|\mathbf{x}\|_\infty)F(\mathbf{y}).$$

**Lemma 7** (Lemma 1 of (Dürr et al., 2019)). *For every two vectors $\mathbf{x}, \mathbf{y} \in [0,1]^d$ and any continuously differentiable non-negative DR-submodular function $F$ we have*

$$\langle \nabla F(\mathbf{x}), \mathbf{y} - \mathbf{x} \rangle \ge F(\mathbf{x} \vee \mathbf{y}) + F(\mathbf{x} \wedge \mathbf{y}) - 2F(\mathbf{x}).$$

## E  OFFLINE

As we mentioned in Section 3.1, when given a value oracle, we solve the problem of DR-submodular maximization over the convex set $\hat{\mathcal{K}}$. However, if $\mathbf{0} \in \mathcal{K}$, then it can be immediately seen from the definition that $\mathbf{0} \notin \hat{\mathcal{K}}$. Here we want to present statements in the offline setting that could also be applied to $\hat{\mathcal{K}}$ as well as $\mathcal{K}$. For this reason, we adapt the categories (A)-(D) described in Section 3 to accommodate more general feasible sets.

(A$'$) The functions $F$ is monotone DR-submodular and $\mathcal{K}$ has a minimum point, i.e. there is a unique point $\underline{\mathbf{u}} \in \mathcal{K}$ such that for all $\mathbf{z} \in \mathcal{K}$, we have $\underline{\mathbf{u}} \leq \mathbf{z}$.

(B$'$) The function $F$ is non-monotone DR-submodular and $\mathcal{K}$ is a downward-closed convex set.

(C$'$) The function $F$ is monotone DR-submodular and $\mathcal{K}$ is a general convex set.

(D$'$) The function $F$ is non-monotone DR-submodular and $\mathcal{K}$ is a general convex set.

Note that since we are considering the offline case here, there is only a single function and not a sequence of functions. Here we use the value for $\alpha$, and the functions update and oracle-adv as before.

**Lemma 8.** *Let $\mathcal{K} \subseteq [0,1]^d$ be a closed convex set and let $F$ be a non-negative $M_1$-Lipschitz $M_2$-smooth continuous DR-submodular function over $\mathcal{K}$ that is bounded by $M_0$. Moreover, assume that the pair $(F, \mathcal{K})$ belong to one of the categories $(A')$-$(D')$ described above. Let $(\mathbf{v}^{(k)})_{k=1}^{K}$ be a sequence of points in $\mathcal{K}$, $\underline{\mathbf{u}} \in \operatorname{argmin}_{\mathbf{x} \in \mathcal{K}} \|\mathbf{x}\|_{\infty}$ and let $(\mathbf{x}^{(k)})_{k=1}^{K+1}$ be a sequence of points generated using the following rule.*

$$\mathbf{x}^{(1)} = \underline{\mathbf{u}}, \quad \forall k \in [K], \ \mathbf{x}^{(k+1)} = \text{update}(\mathbf{x}^{(k)}, \mathbf{v}^{(k)}, \mathbf{x}^{(1)}).$$

*Then, the sequence $(\mathbf{x}^{(k)})_{k=1}^{K+1}$ is contained in $\mathcal{K}$ and for any $\mathbf{x}^* \in \mathcal{K}$, we have*

$$F(\mathbf{x}^{(K+1)}) \geq \alpha F(\mathbf{x}^*) - \Gamma + \sum_{k=1}^{K} \eta^{(k)} \langle \text{oracle-adv}(\nabla F(\mathbf{x}^{(k)}), \mathbf{x}^{(k)}), \mathbf{v}^{(k)} - \mathbf{x}^* \rangle,$$

*where*

$$\Gamma := \begin{cases} \frac{M_2 D^2}{2K} & (A'); \\ \frac{M_2 D^2}{2K} + (M_0 + M_1)\|\underline{\mathbf{u}}\|_{\infty} & (B'); \\ \frac{8M_0 + M_2 D^2 \log(K)^2}{8K} & (C'); \\ \frac{M_0 + 2M_2 D^2}{4K} & (D'). \end{cases} \qquad \eta^{(k)} := \begin{cases} \left(1 - \frac{1}{K}\right)^{K-k} \frac{1}{K} & (A') \ or \ (B'); \\ (1 - 2\varepsilon_C)^{K-k} \varepsilon_C & (C'); \\ (1 - 2\varepsilon_D)^{K-k} \varepsilon_D & (D'), \end{cases}$$

*for all $k \in [K]$.*

**Remark 5.** *Note that we always have $0 \leq \eta^{(k)} \leq 1$ and*

$$\sum_{k=1}^{K} \eta^{(k)} \leq \begin{cases} \log(K) & (C'); \\ 1 & Otherwise. \end{cases}$$

*Proof.* We consider each case separately.

**Case ($A'$):** First we show that the sequence $(\mathbf{x}^{(k)})_{k=1}^{K+1}$ is contained in $\mathcal{K}$. For any $k \in [K]$, we have

$$\mathbf{x}^{(k+1)} = \underline{\mathbf{u}} + \frac{1}{K} \sum_{s=1}^{k} (\mathbf{v}^{(s)} - \underline{\mathbf{u}}) = \frac{1}{K} \left( (K-k)\underline{\mathbf{u}} + \sum_{s=1}^{k} \mathbf{v}^{(s)} \right),$$

which is a convex combination of $K - k$ copies of $\underline{\mathbf{u}}$ and the $k$ points $\mathbf{v}^{(s)}$, for $s \in [k]$. Therefore we have $\mathbf{x}^{(k+1)} \in \mathcal{K}$.

Let $\varepsilon = \frac{1}{K}$. We want to prove that

$$F(\mathbf{x}^{(K+1)}) \geq \left(1 - \frac{1}{e}\right) F(\mathbf{x}^*) - \frac{M_2 D^2}{2K}$$

$$+ \sum_{k=1}^{K} (1 - \varepsilon)^{K-k} \varepsilon \langle \nabla F(\mathbf{x}^{(k)}), \mathbf{v}^{(k)} - \mathbf{x}^* \rangle.$$

Using the fact that $F$ is $M_2$-smooth, we have

$$F(\mathbf{x}^{(k+1)}) - F(\mathbf{x}^{(k)}) \geq \langle \nabla F(\mathbf{x}^{(k)}), \mathbf{x}^{(k+1)} - \mathbf{x}^{(k)} \rangle - \frac{M_2}{2} \|\mathbf{x}^{(k+1)} - \mathbf{x}^{(k)}\|^2$$

$$= \varepsilon \langle \nabla F(\mathbf{x}^{(k)}), \mathbf{v}^{(k)} - \underline{\mathbf{u}} \rangle - \frac{\varepsilon^2 M_2}{2} \|\mathbf{v}^{(k)} - \underline{\mathbf{u}}\|^2$$

$$\geq \varepsilon \langle \nabla F(\mathbf{x}^{(k)}), \mathbf{v}^{(k)} - \underline{\mathbf{u}} \rangle - \frac{\varepsilon^2 M_2 D^2}{2}.$$

Since $\underline{\mathbf{u}}$ is the minimum point of $\mathcal{K}$, we have $\mathbf{x}^{(k)} \geq \underline{\mathbf{u}}$ and $\mathbf{x}^* \geq \underline{\mathbf{u}}$. Therefore we have $\mathbf{x}^* - \underline{\mathbf{u}} \geq \mathbf{0}$ and $\mathbf{x}^* - \underline{\mathbf{u}} \geq \mathbf{x}^* - \mathbf{x}^{(k)}$. Hence $\mathbf{x}^* - \underline{\mathbf{u}} \geq (\mathbf{x}^* - \mathbf{x}^{(k)}) \vee \mathbf{0}$ Using monotonicity of $F$ and its concavity along non-negative directions, we have

$$
\begin{aligned}
\langle \nabla F(\mathbf{x}^{(k)}), \mathbf{x}^* - \underline{\mathbf{u}} \rangle &\geq \langle \nabla F(\mathbf{x}^{(k)}), (\mathbf{x}^* - \mathbf{x}^{(k)}) \vee 0 \rangle \\
&= \langle \nabla F(\mathbf{x}^{(k)}), \mathbf{x}^* \vee \mathbf{x}^{(k)} - \mathbf{x}^{(k)} \rangle \\
&\geq F(\mathbf{x}^* \vee \mathbf{x}^{(k)}) - F(\mathbf{x}^{(k)}) \\
&\geq F(\mathbf{x}^*) - F(\mathbf{x}^{(k)}).
\end{aligned}
$$

Therefore

$$
\begin{aligned}
F(\mathbf{x}^{(k+1)}) &\geq F(\mathbf{x}^{(k)}) + \varepsilon \langle \nabla F(\mathbf{x}^{(k)}), \mathbf{v}^{(k)} \rangle - \frac{\varepsilon^2 M_2 D^2}{2} \\
&\geq (1-\varepsilon) F(\mathbf{x}^{(k)}) + \varepsilon F(\mathbf{x}^*) + \varepsilon \langle \nabla F(\mathbf{x}^{(k)}), \mathbf{v}^{(k)} - \mathbf{x}^* \rangle - \frac{\varepsilon^2 M_2 D^2}{2}.
\end{aligned}
$$

Using this inequality recursively for $1 \leq k \leq K$, we see that

$$
\begin{aligned}
F(\mathbf{x}^{(K+1)}) &\geq (1-\varepsilon)^K F(\mathbf{x}^{(1)}) + \sum_{k=1}^{K} (1-\varepsilon)^{K-k} \varepsilon F(\mathbf{x}^*) - \sum_{k=1}^{K} (1-\varepsilon)^{K-k} \frac{\varepsilon^2 M_2 D^2}{2} \\
&\quad + \sum_{k=1}^{K} (1-\varepsilon)^{K-k} \varepsilon \langle \nabla F(\mathbf{x}^{(k)}), \mathbf{v}^{(k)} - \mathbf{x}^* \rangle \\
&\geq \sum_{k=1}^{K} (1-\varepsilon)^{K-k} \varepsilon F(\mathbf{x}^*) - \sum_{k=1}^{K} \frac{\varepsilon^2 M_2 D^2}{2} \\
&\quad + \sum_{k=1}^{K} (1-\varepsilon)^{K-k} \varepsilon \langle \nabla F(\mathbf{x}^{(k)}), \mathbf{v}^{(k)} - \mathbf{x}^* \rangle \\
&= \left(1 - (1-\varepsilon)^K\right) F(\mathbf{x}^*) - \frac{K \varepsilon^2 M_2 D^2}{2} \\
&\quad + \sum_{k=1}^{K} (1-\varepsilon)^{K-k} \varepsilon \langle \nabla F(\mathbf{x}^{(k)}), \mathbf{v}^{(k)} - \mathbf{x}^* \rangle \\
&\geq \left(1 - \frac{1}{e}\right) F(\mathbf{x}^*) - \frac{M_2 D^2}{2K} \\
&\quad + \sum_{k=1}^{K} (1-\varepsilon)^{K-k} \varepsilon \langle \nabla F(\mathbf{x}^{(k)}), \mathbf{v}^{(k)} - \mathbf{x}^* \rangle,
\end{aligned}
$$

where we used $(1-\varepsilon)^K = (1 - \frac{1}{K})^K \leq \frac{1}{e}$ in the last inequality.

**Case $(B')$:** First we use induction to show that $\mathbf{x}^{(s)} \in \mathcal{K}$ for all $1 \leq s \leq K+1$. The claim is true for $s = 1$ since $\mathbf{x}^{(s)} = \underline{\mathbf{u}} \in \mathcal{K}$. Assume that the claim is true for $1 \leq s \leq k$. We have

$$
\begin{aligned}
\mathbf{x}^{(k+1)} &= \mathbf{x}^{(k)} + \frac{1}{K}(\mathbf{v}^{(k)} - \underline{\mathbf{u}}) \odot (\mathbf{1} - \mathbf{x}^{(k)}) \\
&= \underline{\mathbf{u}} + \frac{1}{K} \sum_{s=1}^{k} (\mathbf{v}^{(s)} - \underline{\mathbf{u}}) \odot (\mathbf{1} - \mathbf{x}^{(s)}) \\
&= \underline{\mathbf{u}} + \frac{1}{K} \sum_{s=1}^{k} (\tilde{\mathbf{v}}^{(s)} - \underline{\mathbf{u}}) \\
&= \frac{1}{K} \left( (K-k)\underline{\mathbf{u}} + \sum_{s=1}^{k} \tilde{\mathbf{v}}^{(s)} \right),
\end{aligned}
$$

where $\tilde{\mathbf{v}}^{(s)} := \underline{\mathbf{u}} + (\mathbf{v}^{(s)} - \underline{\mathbf{u}}) \odot (\mathbf{1} - \mathbf{x}^{(s)})$. According to the induction hypothesis, for $1 \leq s \leq k$, we have $\mathbf{x}^{(s)} \in \mathcal{K}$ which implies that $\mathbf{0} \leq \mathbf{x}^{(s)} \leq \mathbf{1}$. Hence we have

$$\mathbf{0} \leq (\mathbf{v}^{(s)} - \underline{\mathbf{u}}) \odot (\mathbf{1} - \mathbf{x}^{(s)}) \leq \mathbf{v}^{(s)} - \underline{\mathbf{u}}.$$

Therefore $\underline{\mathbf{u}} \leq \tilde{\mathbf{v}}^{(s)} \leq \mathbf{v}^{(s)}$. Since $\mathcal{K}$ is downward-closed, this implies that $\tilde{\mathbf{v}}^{(s)} \in \mathcal{K}$. In other word, $\mathbf{x}^{(k+1)}$ is a convex combination of $K - k$ copies of $\underline{\mathbf{u}}$ and the $k$ points $\tilde{\mathbf{v}}^{(s)}$, for $s \in [k]$. Therefore we have $\mathbf{x}^{(k+1)} \in \mathcal{K}$.

Let $\varepsilon = \frac{1}{K}$. We want to prove that

$$F(\mathbf{x}^{(K+1)}) \geq \frac{1}{e}(1 - \|\underline{\mathbf{u}}\|_\infty)F(\mathbf{x}^*) - \frac{M_2 D^2}{2K} - M_1 \|\underline{\mathbf{u}}\|$$

$$+ \sum_{k=1}^{K} (1 - \varepsilon)^{K-k} \varepsilon \langle \nabla F(\mathbf{x}^{(k)}) \odot (\mathbf{1} - \mathbf{x}^{(k)}), \mathbf{v}^{(k)} - \mathbf{x}^* \rangle.$$

We start by showing that

$$1 - \|\mathbf{x}^{(k)}\|_\infty \geq (1 - \varepsilon)^{k-1}(1 - \|\underline{\mathbf{u}}\|_\infty), \tag{6}$$

for all $1 \leq k \leq K + 1$. We use induction on $n$ to show that for each coordinate $1 \leq i \leq d$, we have $1 - [\mathbf{x}^{(k)}]_i \geq (1 - \varepsilon)^{k-1}(1 - [\underline{\mathbf{u}}]_i)$. The claim is obvious for $k = 1$. Assuming that the inequality is true for $k$, using the fact that $\mathbf{v}_k - \underline{\mathbf{u}} \leq \mathbf{1}$, we have

$$1 - [\mathbf{x}^{(k+1)}]_i = 1 - [\mathbf{x}^{(k)}]_i - \varepsilon[(\mathbf{v}_k - \underline{\mathbf{u}}) \odot (\mathbf{1} - \mathbf{x}^{(k)})]_i$$
$$= 1 - [\mathbf{x}^{(k)}]_i - \varepsilon([\mathbf{v}_k]_i - [\underline{\mathbf{u}}]_i)(1 - [\mathbf{x}^{(k)}]_i)$$
$$\geq 1 - [\mathbf{x}^{(k)}]_i - \varepsilon(1 - [\mathbf{x}^{(k)}]_i)$$
$$= (1 - \varepsilon)(1 - [\mathbf{x}^{(k)}]_i) \geq (1 - \varepsilon)^k(1 - [\underline{\mathbf{u}}]_i),$$

which completes the proof by induction.

$$F(\mathbf{x}^{(k+1)}) - F(\mathbf{x}^{(k)}) \geq \langle \nabla F(\mathbf{x}^{(k)}), \mathbf{x}^{(k+1)} - \mathbf{x}^{(k)} \rangle - \frac{M_2}{2} \|\mathbf{x}^{(k+1)} - \mathbf{x}^{(k)}\|^2$$

$$= \varepsilon \langle \nabla F(\mathbf{x}^{(k)}), (\mathbf{v}^{(k)} - \underline{\mathbf{u}}) \odot (\mathbf{1} - \mathbf{x}^{(k)}) \rangle - \frac{\varepsilon^2 M_2}{2} \|(\mathbf{v}^{(k)} - \underline{\mathbf{u}}) \odot (\mathbf{1} - \mathbf{x}^{(k)})\|^2$$

$$\geq \varepsilon \langle \nabla F(\mathbf{x}^{(k)}), (\mathbf{v}^{(k)} - \underline{\mathbf{u}}) \odot (\mathbf{1} - \mathbf{x}^{(k)}) \rangle - \frac{\varepsilon^2 M_2 D^2}{2}.$$

Since $F$ is DR-submodular, it is concave along non-negative directions. Therefore, using Lemma 6 and Equation equation 6, we have

$$\langle \nabla F(\mathbf{x}^{(k)}), \mathbf{x}^* \odot (\mathbf{1} - \mathbf{x}^{(k)}) \rangle \geq F(\mathbf{x}^{(k)} + \mathbf{x}^* \odot (\mathbf{1} - \mathbf{x}^{(k)})) - F(\mathbf{x}^{(k)})$$
$$\geq (1 - \|\mathbf{x}^{(k)}\|_\infty)F(\mathbf{x}^*) - F(\mathbf{x}^{(k)})$$
$$\geq (1 - \varepsilon)^{k-1}(1 - \|\mathbf{x}^{(1)}\|_\infty)F(\mathbf{x}^*) - F(\mathbf{x}^{(k)}).$$

Therefore

$$F(\mathbf{x}^{(k+1)}) \geq F(\mathbf{x}^{(k)}) + \varepsilon \langle \nabla F(\mathbf{x}^{(k)}), (\mathbf{v}^{(k)} - \underline{\mathbf{u}}) \odot (\mathbf{1} - \mathbf{x}^{(k)}) \rangle - \frac{\varepsilon^2 M_2 D^2}{2}$$

$$\geq (1 - \varepsilon)F(\mathbf{x}^{(k)}) + \varepsilon(1 - \varepsilon)^{k-1}(1 - \|\mathbf{x}^{(1)}\|_\infty)F(\mathbf{x}^*)$$

$$+ \varepsilon \langle \nabla F(\mathbf{x}^{(k)}), (\mathbf{v}^{(k)} - \underline{\mathbf{u}} - \mathbf{x}^*) \odot (\mathbf{1} - \mathbf{x}^{(k)}) \rangle - \frac{\varepsilon^2 M_2 D^2}{2}.$$

Using this inequality recursively for $1 \leq k \leq K$, we see that

$$F(\mathbf{x}^{(K+1)}) \geq (1-\varepsilon)^K F(\mathbf{x}^{(1)}) + \sum_{k=1}^{K}(1-\varepsilon)^{K-1}\varepsilon(1 - \|\mathbf{x}^{(1)}\|_\infty)F(\mathbf{x}^*)$$

$$- \sum_{k=1}^{K}(1-\varepsilon)^{K-k}\frac{\varepsilon^2 M_2 D^2}{2}$$

$$+ \sum_{k=1}^{K}(1-\varepsilon)^{K-k}\varepsilon\langle \nabla F(\mathbf{x}^{(k)}), (\mathbf{v}^{(k)} - \underline{\mathbf{u}} - \mathbf{x}^*) \odot (1 - \mathbf{x}^{(k)})\rangle$$

$$\geq \sum_{k=1}^{K}(1-\varepsilon)^{K-1}\varepsilon(1 - \|\mathbf{x}^{(1)}\|_\infty)F(\mathbf{x}^*) - \sum_{k=1}^{K}\frac{\varepsilon^2 M_2 D^2}{2}$$

$$+ \sum_{k=1}^{K}(1-\varepsilon)^{K-k}\varepsilon\langle \nabla F(\mathbf{x}^{(k)}), (\mathbf{v}^{(k)} - \underline{\mathbf{u}} - \mathbf{x}^*) \odot (1 - \mathbf{x}^{(k)})\rangle$$

$$= \left(1 - \frac{1}{K}\right)^{K-1}(1 - \|\mathbf{x}^{(1)}\|_\infty)F(\mathbf{x}^*) - \frac{M_2 D^2}{2K}$$

$$+ \sum_{k=1}^{K}(1-\varepsilon)^{K-k}\varepsilon\langle \nabla F(\mathbf{x}^{(k)}), (\mathbf{v}^{(k)} - \underline{\mathbf{u}} - \mathbf{x}^*) \odot (1 - \mathbf{x}^{(k)})\rangle$$

$$\geq \frac{1}{e}(1 - \|\mathbf{x}^{(1)}\|_\infty)F(\mathbf{x}^*) - \frac{M_2 D^2}{2K}$$

$$+ \sum_{k=1}^{K}(1-\varepsilon)^{K-k}\varepsilon\langle \nabla F(\mathbf{x}^{(k)}) \odot (1 - \mathbf{x}^{(k)}), \mathbf{v}^{(k)} - \underline{\mathbf{u}} - \mathbf{x}^*\rangle$$

$$= \frac{1}{e}(1 - \|\mathbf{x}^{(1)}\|_\infty)F(\mathbf{x}^*) - \frac{M_2 D^2}{2K}$$

$$- \sum_{k=1}^{K}(1-\varepsilon)^{K-k}\varepsilon\langle \nabla F(\mathbf{x}^{(k)}) \odot (1 - \mathbf{x}^{(k)}), \underline{\mathbf{u}}\rangle$$

$$+ \sum_{k=1}^{K}(1-\varepsilon)^{K-k}\varepsilon\langle \nabla F(\mathbf{x}^{(k)}) \odot (1 - \mathbf{x}^{(k)}), \mathbf{v}^{(k)} - \mathbf{x}^*\rangle,$$

where we used $(1 - \frac{1}{K})^{K-1} \geq \frac{1}{e}$ and the fact that $\langle \mathbf{a} \odot \mathbf{b}, \mathbf{c}\rangle = \langle \mathbf{a}, \mathbf{c} \odot \mathbf{b}\rangle = \sum_{i=1}^{d} a_i b_i c_i$ for all points $\mathbf{a}, \mathbf{b}, \mathbf{c} \in \mathbb{R}^d$ in the last inequality. Finally, we note that

$$\sum_{k=1}^{K}(1-\varepsilon)^{K-k}\varepsilon\langle \nabla F(\mathbf{x}^{(k)}) \odot (1 - \mathbf{x}^{(k)}), \underline{\mathbf{u}}\rangle \leq \sum_{k=1}^{K}(1-\varepsilon)^{K-k}\varepsilon \left\|\nabla F(\mathbf{x}^{(k)}) \odot (1 - \mathbf{x}^{(k)})\right\| \|\underline{\mathbf{u}}\|$$

$$\leq \sum_{k=1}^{K}(1-\varepsilon)^{K-k}\varepsilon M_1 \|\underline{\mathbf{u}}\|$$

$$\leq \sum_{k=1}^{K}\varepsilon M_1 \|\underline{\mathbf{u}}\| = M_1 \|\underline{\mathbf{u}}\| \leq M_1 \|\underline{\mathbf{u}}\|_\infty,$$

and

$$\frac{1}{e}F(\mathbf{x}^*)\|\mathbf{x}^{(1)}\|_\infty \leq M_0 \|\underline{\mathbf{u}}\|_\infty.$$

Therefore

$$F(\mathbf{x}^{(K+1)}) \geq \frac{1}{e}F(\mathbf{x}^*) - \frac{M_2 D^2}{2K} - (M_1 + M_0)\|\underline{\mathbf{u}}\|_\infty$$

$$+ \sum_{k=1}^{K}(1-\varepsilon)^{K-k}\varepsilon\langle \nabla F(\mathbf{x}^{(k)}) \odot (1 - \mathbf{x}^{(k)}), \mathbf{v}^{(k)} - \mathbf{x}^*\rangle.$$

**Case $(C')$:** Since the update rule is convex, the the sequence $(\mathbf{x}^{(k)})_{k=1}^{K+1}$ is contained in $\mathcal{K}$. Let $\varepsilon = \varepsilon_C$. We want to show that

$$F(\mathbf{x}^{(K+1)}) \geq \frac{1}{2}F(\mathbf{x}^*) - \frac{8M_0 + M_2 D^2 \log(K)^2}{8K}$$
$$+ \sum_{k=1}^{K}(1 - 2\varepsilon)^{K-k}\varepsilon\langle\nabla F(\mathbf{x}^{(k)}), \mathbf{v}^{(k)} - \mathbf{x}^*\rangle.$$

Using the fact that $F$ is $M_2$-smooth, we have

$$F(\mathbf{x}^{(k+1)}) - F(\mathbf{x}^{(k)}) \geq \langle\nabla F(\mathbf{x}^{(k)}), \mathbf{x}^{(k+1)} - \mathbf{x}^{(k)}\rangle - \frac{L}{2}\|\mathbf{x}^{(k+1)} - \mathbf{x}^{(k)}\|^2$$
$$= \varepsilon\langle\nabla F(\mathbf{x}^{(k)}), \mathbf{v}^{(k)} - \mathbf{x}^{(k)}\rangle - \frac{\varepsilon^2 L}{2}\|\mathbf{v}^{(k)} - \mathbf{x}^{(k)}\|^2$$
$$\geq \varepsilon\langle\nabla F(\mathbf{x}^{(k)}), \mathbf{v}^{(k)} - \mathbf{x}^{(k)}\rangle - \frac{\varepsilon^2 M_2 D^2}{2}.$$

Using Lemma 7 and the fact that $F$ is monotone, we see that

$$\langle\nabla F(\mathbf{x}^{(k)}), \mathbf{x}^* - \mathbf{x}^{(k)}\rangle \geq F(\mathbf{x}^* \vee \mathbf{x}^{(k)}) + F(\mathbf{x}^* \wedge \mathbf{x}^{(k)}) - 2F(\mathbf{x}^{(k)})$$
$$\geq F(\mathbf{x}^*) + F(\mathbf{x}^* \wedge \mathbf{x}^{(k)}) - 2F(\mathbf{x}^{(k)})$$
$$\geq F(\mathbf{x}^*) - 2F(\mathbf{x}^{(k)}).$$

Therefore

$$F(\mathbf{x}^{(k+1)}) \geq F(\mathbf{x}^{(k)}) + \varepsilon\langle\nabla F(\mathbf{x}^{(k)}), \mathbf{v}^{(k)} - \mathbf{x}^{(k)}\rangle - \frac{\varepsilon^2 M_2 D^2}{2}$$
$$\geq (1 - 2\varepsilon)F(\mathbf{x}^{(k)}) + \varepsilon F(\mathbf{x}^*) + \varepsilon\langle\nabla F(\mathbf{x}^{(k)}), \mathbf{v}^{(k)} - \mathbf{x}^*\rangle - \frac{\varepsilon^2 M_2 D^2}{2}.$$

Using this inequality recursively for $1 \leq k \leq K$, we see that

$$F(\mathbf{x}^{(K+1)}) \geq (1 - 2\varepsilon)^K F(\mathbf{x}^{(1)}) + \sum_{k=1}^{K}(1 - 2\varepsilon)^{K-1}\varepsilon F(\mathbf{x}^*) - \sum_{k=1}^{K}(1 - 2\varepsilon)^{K-k}\frac{\varepsilon^2 M_2 D^2}{2}$$
$$+ \sum_{k=1}^{K}(1 - 2\varepsilon)^{K-k}\varepsilon\langle\nabla F(\mathbf{x}^{(k)}), \mathbf{v}^{(k)} - \mathbf{x}^*\rangle$$
$$\geq \sum_{k=1}^{K}(1 - 2\varepsilon)^{K-1}\varepsilon F(\mathbf{x}^*) - \sum_{k=1}^{K}\frac{\varepsilon^2 M_2 D^2}{2}$$
$$+ \sum_{k=1}^{K}(1 - 2\varepsilon)^{K-k}\varepsilon\langle\nabla F(\mathbf{x}^{(k)}), \mathbf{v}^{(k)} - \mathbf{x}^*\rangle$$
$$= \frac{1}{2}\left(1 - (1 - 2\varepsilon)^K\right)F(\mathbf{x}^*) - \frac{K\varepsilon^2 M_2 D^2}{2}$$
$$+ \sum_{k=1}^{K}(1 - 2\varepsilon)^{K-k}\varepsilon\langle\nabla F(\mathbf{x}^{(k)}), \mathbf{v}^{(k)} - \mathbf{x}^*\rangle$$
$$\geq \frac{1}{2}F(\mathbf{x}^*) - \frac{8M_0 + M_2 D^2 \log(K)^2}{8K}$$
$$+ \sum_{k=1}^{K}(1 - 2\varepsilon)^{K-k}\varepsilon\langle\nabla F(\mathbf{x}^{(k)}), \mathbf{v}^{(k)} - \mathbf{x}^*\rangle,$$

where the last inequality follows from the fact that $F$ is bounded by $M_0$ and

$$(1 - 2\varepsilon)^K = \left(1 - \frac{\log(K)}{K}\right)^K \leq e^{-\log(K)} = \frac{1}{K}.$$

**Case $(D')$:** Since the update rule is convex, the the sequence $(\mathbf{x}^{(k)})_{k=1}^{K+1}$ is contained in $\mathcal{K}$. Let $\varepsilon = \varepsilon_D$. We want to show that

$$F(\mathbf{x}^{(K+1)}) \geq \frac{1}{4}(1 - \|\underline{\mathbf{u}}\|_\infty)F(\mathbf{x}^*) - \frac{M_0 + 2M_2 D^2}{4K}$$
$$+ \sum_{k=1}^{K}(1 - 2\varepsilon)^{K-k}\varepsilon\langle\nabla F(\mathbf{x}^{(k)}), \mathbf{v}^{(k)} - \mathbf{x}^*\rangle.$$

First we prove that

$$1 - \|\mathbf{x}^{(k)}\|_\infty \geq (1 - \varepsilon)^{k-1}(1 - \|\underline{\mathbf{u}}\|_\infty), \tag{7}$$

for all $1 \leq k \leq K + 1$. We use induction on $k$ to show that for each coordinate $1 \leq i \leq d$, we have $1 - [\mathbf{x}^{(k)}]_i \geq (1 - \varepsilon)^{k-1}(1 - [\underline{\mathbf{u}}]_i)$. The claim is obvious for $k = 1$. Assuming that the inequality is true for $k$, we have

$$1 - [\mathbf{x}^{(k+1)}]_i = 1 - (1 - \varepsilon)[\mathbf{x}^{(k)}]_i - \varepsilon[\mathbf{v}^{(k)}]_i \geq 1 - (1 - \varepsilon)[\mathbf{x}^{(k)}]_i - \varepsilon$$
$$= (1 - \varepsilon)(1 - [\mathbf{x}^{(k)}]_i) \geq (1 - \varepsilon)^n(1 - [\underline{\mathbf{u}}]_i),$$

which completes the proof by induction.

Using the fact that $F$ is $L$-smooth, we have

$$F(\mathbf{x}^{(k+1)}) - F(\mathbf{x}^{(k)}) \geq \langle\nabla F(\mathbf{x}^{(k)}), \mathbf{x}^{(k+1)} - \mathbf{x}^{(k)}\rangle - \frac{L}{2}\|\mathbf{x}^{(k+1)} - \mathbf{x}^{(k)}\|^2$$
$$= \varepsilon\langle\nabla F(\mathbf{x}^{(k)}), \mathbf{v}^{(k)} - \mathbf{x}^{(k)}\rangle - \frac{\varepsilon^2 L}{2}\|\mathbf{v}^{(k)} - \mathbf{x}^{(k)}\|^2 \tag{8}$$
$$\geq \varepsilon\langle\nabla F(\mathbf{x}^{(k)}), \mathbf{v}^{(k)} - \mathbf{x}^{(k)}\rangle - \frac{\varepsilon^2 M_2 D^2}{2}.$$

Using non-negativity of $F$, Lemmas 7 and 5 and Equation 7, we see that

$$\langle\nabla F(\mathbf{x}^{(k)}), \mathbf{x}^* - \mathbf{x}^{(k)}\rangle \geq F(\mathbf{x}^* \vee \mathbf{x}^{(k)}) + F(\mathbf{x}^* \wedge \mathbf{x}^{(k)}) - 2F(\mathbf{x}^{(k)})$$
$$\geq F(\mathbf{x}^* \vee \mathbf{x}^{(k)}) - 2F(\mathbf{x}^{(k)})$$
$$\geq (1 - \|\mathbf{x}^{(k)}\|_\infty)F(\mathbf{x}^*) - 2F(\mathbf{x}^{(k)})$$
$$\geq (1 - \varepsilon)^{k-1}(1 - \|\underline{\mathbf{u}}\|_\infty)F(\mathbf{x}^*) - 2F(\mathbf{x}^{(k)}).$$

Therefore

$$F(\mathbf{x}^{(k+1)}) \geq F(\mathbf{x}^{(k)}) + \varepsilon\langle\nabla F(\mathbf{x}^{(k)}), \mathbf{v}^{(k)} - \mathbf{x}^{(k)}\rangle - \frac{\varepsilon^2 M_2 D^2}{2}$$
$$\geq (1 - 2\varepsilon)F(\mathbf{x}^{(k)}) + \varepsilon(1 - \varepsilon)^{k-1}(1 - \|\underline{\mathbf{u}}\|_\infty)F(\mathbf{x}^*)$$
$$+ \varepsilon\langle\nabla F(\mathbf{x}^{(k)}), \mathbf{v}^{(k)} - \mathbf{x}^*\rangle - \frac{\varepsilon^2 M_2 D^2}{2}.$$

Using this inequality recursively for $1 \leq k \leq K$, we see that

$$F(\mathbf{x}^{(K+1)}) \geq (1 - 2\varepsilon)^K F(\mathbf{x}^{(1)}) + \sum_{k=1}^{K}(1 - 2\varepsilon)^{K-1}\varepsilon(1 - \varepsilon)^{k-1}(1 - \|\underline{\mathbf{u}}\|_\infty)F(\mathbf{x}^*)$$
$$- \left(\sum_{k=1}^{K}(1 - 2\varepsilon)^{K-k}\right)\frac{\varepsilon^2 M_2 D^2}{2} + \sum_{k=1}^{K}(1 - 2\varepsilon)^{K-k}\varepsilon\langle\nabla F(\mathbf{x}^{(k)}), \mathbf{v}^{(k)} - \mathbf{x}^*\rangle$$
$$\geq \left(\sum_{k=1}^{K}(1 - 2\varepsilon)^{K-1}\varepsilon(1 - \varepsilon)^{k-1}\right)(1 - \|\underline{\mathbf{u}}\|_\infty)F(\mathbf{x}^*)$$
$$- \frac{K\varepsilon^2 M_2 D^2}{2} + \sum_{k=1}^{K}(1 - 2\varepsilon)^{K-k}\varepsilon\langle\nabla F(\mathbf{x}^{(k)}), \mathbf{v}^{(k)} - \mathbf{x}^*\rangle.$$

We have $(1 + \frac{c}{N}) \geq e^c(1 - \frac{c^2}{2N})$ for $c \geq 0$ and $N \geq 1$. [3] Therefore

$$\varepsilon \sum_{k=1}^{K}(1-2\varepsilon)^{K-k}(1-\varepsilon)^{k-1} = (1-2\varepsilon)^{K-1}\left((1+\varepsilon)^K - 1\right)$$

$$\geq e^{-2\log(2)}\left(\left(1 + \frac{\log(2)}{K}\right)^K - 1\right)$$

$$\geq e^{-2\log(2)}\left(e^{\log 2}\left(1 - \frac{\log(2)^2}{2K}\right) - 1\right)$$

$$= \frac{1}{4}\left(1 - \frac{\log(2)^2}{K}\right)$$

$$\geq \frac{1}{4} - \frac{1}{4K}.$$

Hence

$$F(\mathbf{x}^{(K+1)}) \geq \left(\sum_{k=1}^{K}(1-2\varepsilon)^{K-1}\varepsilon(1-\varepsilon)^{k-1}\right)(1 - \|\underline{\mathbf{u}}\|_\infty)F(\mathbf{x}^*)$$

$$-\frac{\log(2)^2 M_2 D^2}{2K} + \sum_{k=1}^{K}(1-2\varepsilon)^{K-k}\varepsilon\langle\nabla F(\mathbf{x}^{(k)}), \mathbf{v}^{(k)} - \mathbf{x}^*\rangle$$

$$\geq \left(\frac{1}{4} - \frac{1}{4K}\right)(1 - \|\underline{\mathbf{u}}\|_\infty)F(\mathbf{x}^*)$$

$$-\frac{\log(2)^2 M_2 D^2}{2K} + \sum_{k=1}^{K}(1-2\varepsilon)^{K-k}\varepsilon\langle\nabla F(\mathbf{x}^{(k)}), \mathbf{v}^{(k)} - \mathbf{x}^*\rangle$$

$$\geq \frac{1}{4}(1 - \|\underline{\mathbf{u}}\|_\infty)F(\mathbf{x}^*)$$

$$-\frac{M_0 + 2M_2 D^2}{4K} + \sum_{k=1}^{K}(1-2\varepsilon)^{K-k}\varepsilon\langle\nabla F(\mathbf{x}^{(k)}), \mathbf{v}^{(k)} - \mathbf{x}^*\rangle. \qquad \blacksquare$$

## F  PROOF OF THEOREM 1

*Proof.* The key idea of Algorithm 1 is to use the average of several functions in a certain group (*e.g.*, a block) to represent the functions. Note the regret is calculated by the sum of all the reward functions, and the sum of average functions is exactly the sum of all the functions divided by the block size, so we can use the average function to analyze the regret. Let

$$\bar{F}_q(\mathbf{x}) := \frac{1}{L}\sum_{l=1}^{L}\hat{F}_{t_{q,l}}(\mathbf{x}), \qquad \bar{\hat{F}}_q(\mathbf{x}) := \frac{1}{L}\sum_{l=1}^{L}\hat{F}_{t_{q,l}}(\mathbf{x}),$$

denote the average of functions in block $q$ and the average of smoothed version of the functions in block $q$ respectively.

Let $\mathbf{y}^* = \arg\max_{\mathbf{y}\in\mathcal{K}}\sum_{t=1}^{T}F_t(\mathbf{y})$ and $\hat{\mathbf{y}}^* = \arg\max_{\mathbf{y}\in\hat{\mathcal{K}}}\sum_{t=1}^{T}F_t(\mathbf{y})$. Note that both $\mathbf{y}^*$ and $\hat{\mathbf{y}}^*$ maximize the same expression, but over different domains. Using the definition of the $\alpha$-regret and

---

[3]For $\mathbf{x} \geq 0$, we have $\log(1 + \mathbf{x}) \geq \mathbf{x} - \frac{\mathbf{x}^2}{2}$ and $-\mathbf{x} \geq \log(1 - \mathbf{x})$. Therefore $N\log(1 + \frac{c}{N}) \geq N(\frac{c}{N} - \frac{c^2}{2N^2}) = c - \frac{c^2}{2N} \geq c + \log(1 - \frac{c^2}{2N})$.

Lemma 1, we have

$$
\begin{aligned}
\mathcal{R}_\alpha &= \sum_{t=1}^{T} \left( \alpha F_t(\mathbf{y}^*) - F_t(\mathbf{y}_t) \right) \\
&= \sum_{t=1}^{T} \left( \hat{F}_t(\mathbf{y}_t) - F_t(\mathbf{y}_t) \right) + \sum_{t=1}^{T} \alpha \left( F_t(\hat{\mathbf{y}}^*) - \hat{F}_t(\hat{\mathbf{y}}^*) \right) + \sum_{t=1}^{T} \alpha \left( F_t(\mathbf{y}^*) - F_t(\hat{\mathbf{y}}^*) \right) \\
&\quad + \sum_{t=1}^{T} \left( \alpha \hat{F}_t(\hat{\mathbf{y}}^*) - \hat{F}_t(\mathbf{y}_t) \right) \\
&\leq 2M_1 T \delta + \sum_{t=1}^{T} \alpha \left( F_t(\mathbf{y}^*) - F_t(\hat{\mathbf{y}}^*) \right) + \sum_{t=1}^{T} \left( \alpha \hat{F}_t(\hat{\mathbf{y}}^*) - \hat{F}_t(\mathbf{y}_t) \right).
\end{aligned}
$$

According to Lemma 4, there exists $\mathbf{y}' \in \hat{\mathcal{K}}$ such that $\|\mathbf{y}^* - \mathbf{y}'\| \leq \delta' = \frac{\delta D}{r}$. Therefore

$$
\begin{aligned}
\sum_{t=1}^{T} \left( F_t(\mathbf{y}^*) - F_t(\hat{\mathbf{y}}^*) \right) &= \sum_{t=1}^{T} \left( F_t(\mathbf{y}^*) - F_t(\mathbf{y}') \right) + \sum_{t=1}^{T} \left( F_t(\mathbf{y}') - F_t(\hat{\mathbf{y}}^*) \right) \\
&\leq \sum_{t=1}^{T} M_1 \|\mathbf{y}^* - \mathbf{y}'\| + 0 \\
&\leq M_1 T \delta'.
\end{aligned}
$$

Hence we have

$$
\begin{aligned}
\mathcal{R}_\alpha &\leq 2M_1 T \delta + \sum_{t=1}^{T} \alpha \left( F_t(\mathbf{y}^*) - F_t(\hat{\mathbf{y}}^*) \right) + \sum_{t=1}^{T} \left( \alpha \hat{F}_t(\hat{\mathbf{y}}^*) - \hat{F}_t(\mathbf{y}_t) \right) \\
&\leq \left( 2 + \frac{D}{r} \right) M_1 T \delta + \sum_{t=1}^{T} \left( \alpha \hat{F}_t(\hat{\mathbf{y}}^*) - \hat{F}_t(\mathbf{y}_t) \right) \qquad (9) \\
&= \left( 2 + \frac{D}{r} \right) M_1 T \delta + L \sum_{q=1}^{Q} \left( \frac{\alpha}{L} \sum_{l=1}^{L} \hat{F}_{t_{q,l}}(\hat{\mathbf{y}}^*) - \frac{1}{L} \sum_{l=1}^{L} \hat{F}_{t_{q,l}}(\mathbf{y}_{t_{q,l}}) \right) \\
&= \left( 2 + \frac{D}{r} \right) M_1 T \delta + L \sum_{q=1}^{Q} \left( \frac{\alpha}{L} \sum_{l=1}^{L} \hat{F}_{t_{q,l}}(\hat{\mathbf{y}}^*) - \frac{1}{L} \sum_{l=1}^{L} \hat{F}_{t_{q,l}}(\mathbf{x}_q) \right) \\
&= \left( 2 + \frac{D}{r} \right) M_1 T \delta + L \sum_{q=1}^{Q} \left( \alpha \bar{\hat{F}}_q(\hat{\mathbf{y}}^*) - \bar{\hat{F}}_q(\mathbf{x}_q) \right). \qquad (10)
\end{aligned}
$$

Recall that $(t_{q,1}, \ldots, t_{q,L})$ is a random permutation of $\{(q-1)L+1, \cdots, qL\}$, thus $F_{t_{q,l}}(\mathbf{x})$ is a random function and we have $\mathbb{E}[F_{t_{q,l}}(\mathbf{x})] = \bar{F}_q(\mathbf{x})$. Therefore, we see that $\mathbb{E}[\tilde{F}_{t_{q,l}}(\mathbf{x})] = \bar{F}_q(\mathbf{x})$ and $\mathbb{E}[\tilde{\nabla} F_{t_{q,l}}(\mathbf{x})] = \nabla \bar{F}_q(\mathbf{x})$. Define $\mathcal{F}_q$ to be the $\sigma$-algebra generated by all of the random variables in blocks $1, \cdots, q-1$ and $(\mathbf{v}_q^{(k)})_{k=1}^{K}$. Then, conditioned on $\mathcal{F}_q$, the values of $\mathbf{v}_q^{(k)}$ and $\mathbf{x}_q^{(k)}$ are deterministic. If we have access to stochastic gradient oracles, using the definitions of $\mathbf{d}_q^{(k)}$ and stochastic gradient oracles, we have

$$
\mathbf{d}_q^{(k)} = \tilde{\nabla} F_{t_{q,l}}(\mathbf{x}_q^{(k)}) = \tilde{\nabla} F_{t_{q,l}}(\mathbf{x}_q^{(k)}, \mathbf{z}_q^{(k)}),
$$

where $\mathbf{z}_q^{(k)}$ is a random variable distributed according to $p_1^{t_{q,l}}(\cdot; \mathbf{x}_q^{(k)})$. Hence we have

$$
\begin{aligned}
\mathbb{E}\left[\mathbf{d}_q^{(k)}|\mathcal{F}_q\right] &= \mathbb{E}\left[\tilde{\nabla} F_{t_{q,l}}(\mathbf{x}_q^{(k)}, \mathbf{z}_q^{(k)})|\mathcal{F}_q\right] \\
&= \mathbb{E}\left[\mathbb{E}\left[\tilde{\nabla} F_{t_{q,l}}(\mathbf{x}_q^{(k)}, \mathbf{z}_q^{(k)})|t_{q,l}\right]|\mathcal{F}_q\right] \\
&= \mathbb{E}\left[\nabla F_{t_{q,l}}(\mathbf{x}_q^{(k)})|\mathcal{F}_q\right] \\
&= \nabla \bar{F}_q(\mathbf{x}_q^{(k)}) \\
&= \nabla \hat{\bar{F}}_q(\mathbf{x}_q^{(k)}),
\end{aligned}
$$

where the third equality follows from the unbiasedness of the gradient oracle and the last equality follows from the fact that when we are given a gradient oracle, we have $\delta = 0$ and $\hat{F}_t = F_t$. Similarly, if we have access to stochastic value oracles, using the definition of $\mathbf{d}_q^{(k)}$ and stochastic value oracles, we have

$$
\mathbf{d}_q^{(k)} = \frac{d'}{\delta} \tilde{F}_{t_{q,l}}\left(\mathbf{x}_q^{(k)} + \delta \mathbf{u}_q^{(k)}\right) \mathbf{u}_q^{(k)} = \frac{d'}{\delta} \tilde{F}_{t_{q,l}}\left(\mathbf{x}_q^{(k)} + \delta \mathbf{u}_q^{(k)}, \mathbf{z}_q^{(k)}\right) \mathbf{u}_q^{(k)},
$$

where $\mathbf{z}_q^{(k)}$ is a random variable distributed according to $p_0^{t_{q,l}}(\cdot; \mathbf{x}_q^{(k)} + \delta \mathbf{u}_q^{(k)})$. Hence we have

$$
\begin{aligned}
\mathbb{E}\left[\mathbf{d}_q^{(k)}|\mathcal{F}_q\right] &= \mathbb{E}\left[\frac{d'}{\delta} \tilde{F}_{t_{q,l}}\left(\mathbf{x}_q^{(k)} + \delta \mathbf{u}_q^{(k)}, \mathbf{z}_q^{(k)}\right) \mathbf{u}_q^{(k)}|\mathcal{F}_q\right] \\
&= \mathbb{E}\left[\mathbb{E}\left[\frac{d'}{\delta} \tilde{F}_{t_{q,l}}\left(\mathbf{x}_q^{(k)} + \delta \mathbf{u}_q^{(k)}, \mathbf{z}_q^{(k)}\right) \mathbf{u}_q^{(k)}|t_{q,l}\right]|\mathcal{F}_q\right] \\
&= \mathbb{E}\left[\mathbb{E}\left[\mathbb{E}\left[\frac{d'}{\delta} \tilde{F}_{t_{q,l}}\left(\mathbf{x}_q^{(k)} + \delta \mathbf{u}_q^{(k)}, \mathbf{z}_q^{(k)}\right) \mathbf{u}_q^{(k)}|\mathbf{u}_q^{(k)}\right]|t_{q,l}\right]|\mathcal{F}_q\right] \\
&= \mathbb{E}\left[\mathbb{E}\left[\frac{d'}{\delta} F_{t_{q,l}}\left(\mathbf{x}_q^{(k)} + \delta \mathbf{u}_q^{(k)}\right) \mathbf{u}_q^{(k)}|t_{q,l}\right]|\mathcal{F}_q\right] \\
&= \mathbb{E}\left[\nabla \hat{F}_{t_{q,l}}(\mathbf{x}_q^{(k)})|\mathcal{F}_q\right] \\
&= \nabla \hat{\bar{F}}_q(\mathbf{x}_q^{(k)}),
\end{aligned}
$$

where the fourth equality follows from the unbiasedness of the value oracle, the fifth equality follows from Lemma 2.

Therefore, given any type of oracle, we always have $\mathbb{E}[\mathbf{d}_q^{(k)}|\mathcal{F}_q] = \nabla \hat{\bar{F}}_q(\mathbf{x}_q^{(k)})$, which implies that

$$
\mathbb{E}[\mathbf{g}_q^{(k)}|\mathcal{F}_q] = \text{oracle-adv}(\nabla \hat{\bar{F}}_q(\mathbf{x}_q^{(k)}), \mathbf{x}_q^{(k)}).
$$

Hence

$$
\begin{aligned}
\mathbb{E}[\langle \text{oracle-adv}(\nabla \hat{\bar{F}}_q(\mathbf{x}_q^{(k)}), \mathbf{x}_q^{(k)}), \mathbf{v}_q^{(k)} - \mathbf{x}_q^*\rangle] &= \mathbb{E}\left[\langle \mathbb{E}[\mathbf{g}_q^{(k)}|\mathcal{F}_q], \mathbf{v}_q^{(k)} - \mathbf{x}_q^*\rangle\right] \\
&= \mathbb{E}\left[\mathbb{E}[\langle \mathbf{g}_q^{(k)}, \mathbf{v}_q^{(k)} - \mathbf{x}_q^*\rangle|\mathcal{F}_q]\right] \qquad (11) \\
&= \mathbb{E}[\langle \mathbf{g}_q^{(k)}, \mathbf{v}_q^{(k)} - \mathbf{x}_q^*\rangle].
\end{aligned}
$$

Each $F_t : \mathcal{K} \to \mathbb{R}$ is a non-negative $M_1$-Lipschitz $M_2$-smooth continuous DR-submodular function that is bounded by $M_0$. Therefore, according to Lemma 1, so is $\hat{\bar{F}}_q : \hat{\mathcal{K}} \to \mathbb{R}$. Therefore, using Lemma 8, we have

$$
\hat{\bar{F}}_q(\mathbf{x}_q^{(K+1)}) \geq \alpha \hat{\bar{F}}_q(\hat{\mathbf{y}}^*) - \Gamma + \sum_{k=1}^K \eta^{(k)} \langle \nabla \hat{\bar{F}}_q(\mathbf{x}_q^{(k)}), \mathbf{v}_q^{(k)} - \hat{\mathbf{y}}^*\rangle,
$$

Using Equation 11, we have

$$\mathbb{E}[\bar{\hat{F}}_q(\mathbf{x}_q^{(K+1)})] \geq \alpha \bar{\hat{F}}_q(\hat{\mathbf{y}}^*) - \Gamma + \sum_{k=1}^{K} \eta^{(k)} \mathbb{E}[\langle \mathbf{g}_q^{(k)}, \mathbf{v}_q^{(k)} - \hat{\mathbf{y}}^* \rangle].$$

Plugging this into 10 and using Remark 4, we see that

$$\mathbb{E}[\mathcal{R}_\alpha] \leq \left(2 + \frac{D}{r}\right) M_1 T \delta + L \sum_{q=1}^{Q} \mathbb{E}\left[\alpha \bar{\hat{F}}_q(\hat{\mathbf{y}}^*) - \bar{\hat{F}}_q(\mathbf{x}_q)\right]$$

$$\leq \left(2 + \frac{D}{r}\right) M_1 T \delta + L \sum_{q=1}^{Q} \left(\Gamma - \sum_{k=1}^{K} \eta^{(k)} \mathbb{E}[\langle \mathbf{g}_q^{(k)}, \mathbf{v}_q^{(k)} - \hat{\mathbf{y}}^* \rangle]\right)$$

$$= \left(2 + \frac{D}{r}\right) M_1 T \delta + \Gamma L Q + L \sum_{k=1}^{K} \eta^{(k)} \sum_{q=1}^{Q} \mathbb{E}[\langle \mathbf{g}_q^{(k)}, \hat{\mathbf{y}}^* - \mathbf{v}_q^{(k)} \rangle]$$

$$\leq \left(2 + \frac{D}{r}\right) M_1 T \delta + \Gamma T + L \sum_{k=1}^{K} \eta^{(k)} C D B \sqrt{Q}.$$

Using the definition of $\Gamma$ and Lemma 3, for case $(B)$ we have

$$\left(2 + \frac{D}{r}\right) M_1 T \delta + \Gamma T \leq \left(2 + \frac{D}{r}\right) M_1 T \delta + (M_0 + 2M_2 D^2) \frac{T}{4K} + (M_0 + M_1) \|\underline{\mathbf{u}}\|_\infty T$$

$$\leq \left(2 + \frac{D}{r}\right) M_1 T \delta + (M_0 + 2M_2 D^2) \frac{T}{4K} + (M_0 + M_1) T \frac{\delta}{r}$$

$$\leq \left(M_0 + (3 + D) M_1\right) \frac{T \delta}{r} + (M_0 + 2M_2 D^2) \frac{T}{4K},$$

for cases $(A)$ and $(D)$, we have

$$\left(2 + \frac{D}{r}\right) M_1 T \delta + \Gamma T \leq \left(2 + \frac{D}{r}\right) M_1 T \delta + (M_0 + 2M_2 D^2) \frac{T}{4K}$$

$$\leq \left(M_0 + (3 + D) M_1\right) \frac{T \delta}{r} + (M_0 + 2M_2 D^2) \frac{T}{4K},$$

and for case $(C)$, we have

$$\left(2 + \frac{D}{r}\right) M_1 T \delta + \Gamma T \leq \left(M_0 + (3 + D) M_1\right) \frac{T \delta}{r} + (8M_0 + M_2 D^2 \log(K)^2) \frac{T}{8K}.$$

On the other hand, following Remark 5, we have

$$L \sum_{k=1}^{K} \eta^{(k)} C D B \sqrt{Q} \leq L C D B \sqrt{Q} \cdot \begin{cases} \log(K) & (C'); \\ 1 & \text{Otherwise.} \end{cases}$$

Putting these bounds together, we see that

$$\mathbb{E}[\mathcal{R}_\alpha] \leq \left(2 + \frac{D}{r}\right) M_1 T \delta + \Gamma T + L \sum_{k=1}^{K} \eta^{(k)} C D B \sqrt{Q}$$

$$\leq \left(M_0 + (3 + D) M_1\right) \frac{T \delta}{r} + \begin{cases} (8M_0 + M_2 D^2 \log(K)^2) \frac{T}{8K} + L C D B \sqrt{Q} \log(K) & (C'); \\ (M_0 + 2M_2 D^2) \frac{T}{4K} + L C D B \sqrt{Q} & \text{Otherwise.} \end{cases}$$

$$= \mathcal{R}^u. \qquad \blacksquare$$

## G  PROOF OF THEOREM 2

*Proof.* Let $\mathbf{y}^* = \arg\max_{\mathbf{y} \in \mathcal{K}} \sum_{t=1}^{T} F_t(\mathbf{y})$ and $\hat{\mathbf{y}}^* = \arg\max_{\mathbf{y} \in \hat{\mathcal{K}}} \sum_{t=1}^{T} F_t(\mathbf{y})$ as in the proof of Theorem 1. With the same argument as the proof of Equation 9, we see that

$$\mathcal{R}_\alpha \leq \left(2 + \frac{D}{r}\right) M_1 T \delta + \sum_{t=1}^{T} \left(\alpha \hat{F}_t(\hat{\mathbf{y}}^*) - \hat{F}_t(\mathbf{y}_t)\right).$$

On the other hand, using Lemma 1, we see that $\hat{F}$ is bounded by $M_0$. Therefore we have

$$
\begin{aligned}
\sum_{t=1}^{T} \left( \alpha \hat{F}_t(\hat{\mathbf{y}}^*) - \hat{F}_t(\mathbf{y}_q) \right) &= \sum_{q=1}^{Q} \sum_{k=1}^{K} \left( \hat{F}_{t_{q,k}}(\mathbf{x}_q) - \hat{F}_{t_{q,k}}(\mathbf{y}_{t_{q,k}}) \right) + \sum_{q=1}^{Q} \sum_{l=1}^{L} \left( \alpha \hat{F}_{t_{q,l}}(\hat{\mathbf{y}}^*)) - \hat{F}_{t_{q,l}}(\mathbf{x}_q) \right) \\
&\leq \sum_{q=1}^{Q} \sum_{k=1}^{K} 2M_0 + \sum_{q=1}^{Q} \sum_{l=1}^{L} \left( \alpha \hat{F}_{t_{q,l}}(\hat{\mathbf{y}}^*)) - \hat{F}_{t_{q,l}}(\mathbf{x}_q) \right) \\
&= 2M_0 QK + \sum_{q=1}^{Q} \sum_{l=1}^{L} \left( \alpha \hat{F}_{t_{q,l}}(\hat{\mathbf{y}}^*)) - \hat{F}_{t_{q,l}}(\mathbf{x}_q) \right) \\
&= 2M_0 QK + L \sum_{q=1}^{Q} \left( \alpha \bar{\hat{F}}_q(\hat{\mathbf{y}}^*)) - \bar{\hat{F}}_q(\mathbf{x}_q) \right),
\end{aligned}
$$

where $\bar{F}_q$ and $\bar{\hat{F}}_q$ are defined as in the proof of Theorem 1. Hence

$$
\mathcal{R}_\alpha \leq 2M_0 QK + \left( 2 + \frac{D}{r} \right) M_1 T\delta + L \sum_{q=1}^{Q} \left( \alpha \bar{\hat{F}}_q(\hat{\mathbf{y}}^*)) - \bar{\hat{F}}_q(\mathbf{x}_q) \right).
$$

The update rule for $\mathbf{x}_q$ in Algorithm 2 is the same as Algorithm 1. Therefore, the same arguments in the proof of Theorem 1 to bound $\mathbb{E}\left[ \sum_{q=1}^{Q} \left( \alpha \bar{\hat{F}}_q(\hat{\mathbf{y}}^*)) - \bar{\hat{F}}_q(\mathbf{x}_q) \right) \right]$ can be directly used here without any modifications. Similarly, since the update rules of $\mathbf{x}_q^{(k)}$ are the same in both algorithms, the proof of Equation 11 applies verbatim. Hence we see that

$$
\begin{aligned}
\mathbb{E}[\mathcal{R}_\alpha] &\leq 2M_0 QK + \left( 2 + \frac{D}{r} \right) M_1 T\delta + L \sum_{q=1}^{Q} \mathbb{E}\left[ \alpha \bar{\hat{F}}_q(\hat{\mathbf{y}}^*) - \bar{\hat{F}}_q(\mathbf{x}_q) \right] \\
&\leq 2M_0 QK + \left( 2 + \frac{D}{r} \right) M_1 T\delta + \Gamma T + L \sum_{k=1}^{K} \eta^{(k)} C D B \sqrt{Q} \\
&\leq 2M_0 QK + \mathcal{R}^u.
\end{aligned}
$$

$\blacksquare$

# H EXPERIMENTS - EXTENDED DESCRIPTION

## H.1 EXPERIMENT SETUP

We next test our algorithms for online continuous DR-submodular maximization in the setting of non-monotone objectives, a downward-closed feasible region, and full-information and semi-bandit gradient feedback. Specifically, we use online non-convex/non-concave quadratic maximization. Following (Bian et al., 2017a; Chen et al., 2018b; Zhang et al., 2023), for each round $t$ we randomly generate a quadratic function $F_t(\mathbf{x}) = \frac{1}{2}\mathbf{x}^\top \mathbf{H}\mathbf{x} + \mathbf{h}^\top \mathbf{x} + c$. Here, the coefficient matrix $\mathbf{H}$ is a symmetric matrix whose entries $H_{i,j}$ are drawn from a uniform distribution in $[-10, 0]$. Note that this matrix is the Hessian of $F_t$, so having negative entries ensures $F_t$ is DR-submodular. To make $F_t$ non-monotone, we set $\mathbf{h} = -0.1 \times \mathbf{H}^\top \mathbf{u}$, where $\mathbf{u} \in \mathbb{R}^n$. Similar to (Zhang et al., 2023), we set $\mathbf{u} = \mathbf{1}$. In addition, to ensure non-negativity, the constant term is set to $c = -0.5 * \sum_{i,j} H_{i,j}$.

To form a downward-closed feasible region, we generate a coefficient matrix $\mathbf{A}$ for linear constraints using random samples from the uniform distribution over $[0, 1]$, setting $\mathcal{K} = \{\mathbf{x} \in \mathbb{R}^n | \mathbf{A}\mathbf{x} \leq \mathbf{1}, \mathbf{0} \leq \mathbf{x} \leq \mathbf{1}\}$. Like Zhang et al. (2023), we considered three pairs $(n, m)$ of dimensions $n$ and number of constraints $m$, $\{(25, 15), (40, 20), (50, 50)\}$. For a gradient query $\nabla F_t(\mathbf{x})$, we generate a random noise vector $\mathbf{n}_t \sim \mathcal{N}(\mathbf{0}, \mathbf{1})$ and return the noisy gradient $\widetilde{\nabla} F_t(\mathbf{x}) = \nabla F_t(\mathbf{x}) + \epsilon \frac{\mathbf{n}_t}{\|\mathbf{n}_t\|}$, with a noise scale $\epsilon = 0.1$. For the online linear maximization oracles, for simplicity we use the same gradient ascent based oracles as the experiments in (Zhang et al., 2023).

We vary horizons between $T = 20$ and $T = 500$. For each problem instance ($T$ objective functions and specified feasible region), we first obtain $(1/e)$-approximate solutions $\{\mathbf{x}_t^*\}$ for the running

| $\frac{1}{e}$-regret | Algorithm | n=25, m=15 | n=40, m=20 | n=50, m=50 |
|---|---|---|---|---|
| $T^{1/2}$ | Meta($\beta = 3/2$) | $225.43 \pm 7.62$ | $598.11 \pm 45.39$ | $872.00 \pm 56.26$ |
| | GMFW($\beta = 1/2$) | $2.28 \pm 0.24$ | $5.97 \pm \ \ 0.84$ | $9.41 \pm \ \ 2.88$ |
| $T^{2/3}$ | Meta($\beta = 1$) | $22.27 \pm 1.67$ | $59.83 \pm \ \ 6.19$ | $89.57 \pm \ \ 9.45$ |
| | GMFW($\beta = 0$) | $0.25 \pm 0.03$ | $0.63 \pm \ \ 0.24$ | $0.82 \pm \ \ 0.14$ |
| $T^{3/4}$ | Meta($\beta = 3/4$) | $7.49 \pm 0.65$ | $17.88 \pm \ \ 1.94$ | $26.11 \pm \ \ 3.27$ |
| | ODC | $21.92 \pm 2.19$ | $37.78 \pm \ \ 4.71$ | $55.59 \pm \ \ 6.43$ |
| | SBFW (semi) | $0.07 \pm 0.01$ | $0.17 \pm \ \ 0.03$ | $0.30 \pm \ \ 0.18$ |
| $T^{4/5}$ | Mono | $0.23 \pm 0.03$ | $0.49 \pm \ \ 0.04$ | $0.92 \pm \ \ 0.29$ |

Table 3: Mean values and standard deviations of run times in seconds over ten runs for a horizon $T = 100$. For each regret bound, our full information **GMFW** and semi-bandit **SBFW** have the lowest run-times. The run-time differences are primarily due to the number of online linear maximization oracle updates each round (which increase with $\beta$).

sum $\sum_{t'=1}^{t} F_t(\mathbf{x})$ using an offline algorithm (Bian et al., 2017a). Note that in these experiments, the empirical regret is not based on $1/e$ times the value of the optimal solution, but instead against the value achieved by a near-optimal solution (obtained from a $1/e$-approximation algorithm), a more challenging baseline. Following that, we run three online algorithms from prior works: **ODC** from (Thang & Srivastav, 2021), **Mono** (full information single query) from (Zhang et al., 2023), and **Meta**($\beta$) from (Zhang et al., 2023) for $\beta = \{3/4, 1, 3/2\}$; here and in the following we only explicitly mention the query parameter $\beta$ so that there are $T^\beta$ queries per round and other algorithm parameters are left implicit. We ran our Algorithm 1 (**GMFW**($\beta$) for short) with query parameter $\beta = \{0, 1/4, 1/2\}$ and our semi-bandit Algorithm 2 (**SBFW** for short). (Fig. 1 depicts regret bound and query complexity trade offs for full-information methods.) For each algorithm and experiment setup, we compute the running average cumulative regret $(\sum_{t'=1}^{t} F_t(\mathbf{x}_t^*) - \sum_{t'=1}^{t} F_t(\mathbf{y}_{t'}))/t$.

Figure 2 shows these regrets plotted across time and averaged over 10 independent runs. Mean values and standard deviations are shown. Curves corresponding to our method are depicted with solid lines. Curves are color-coded to match regret bound horizon dependence. For example, our **GMFW**($\beta = 1/2$) and **Meta**($\beta = 3/2$) both have $\tilde{O}(\sqrt{T})$ dependence and are both colored black. We note, however, that the number of queries used and the per-round computational complexity can vary significantly between methods with similar regret bounds. Our methods are generally faster and use significantly fewer queries. Average run-times for a horizon of $T = 100$ are displayed in Table 3. Major differences in run-times are in large part due to the number of online linear maximization oracles used, which in turn is related to the number of per-round queries.

For the experiments we ran to generate Fig. 2, we used a compute cluster. The nodes had AMD EPYC 7543P processors. Individual jobs used 8 cores and 4 GB of memory. For the running times reported in Table 3, we conducted the experiments on a single desktop machine with a 12-core AMD Ryzen Threadripper 1920X processor and 64GB of memory. We used Python (v3.8.10) and CVXOPT (v1.3.0), a free software package for convex optimization.

## H.2 RESULTS AND DISCUSSION

The relative performances of our methods (solid lines) and those from prior works (dashed and dotted lines) do not vary dramatically across the different numbers of dimensions $n$ and numbers of constraints $m$ tested. In each experiment, our **GMFW**($\beta = 1/2$) (black solid line) performs the best. It has the same empirical regret as **Meta**($\beta$) baselines for small horizons, but for larger horizons and especially for larger dimensions $n$, our **GMFW**($\beta = 1/2$) performs better than even **Meta**($\beta = 3/2$) despite using significantly fewer gradient queries and significantly less computation than than **Meta**($\beta = 3/2$) does. As shown in Table 3, for a horizon of $T = 100$ with dimension $n = 50$ and $m = 50$ constraints, on average our **GMFW**($\beta = 1/2$) ran in less than 10 seconds while **Meta**($\beta = 3/2$) took over 14 minutes.

Our **GMFW**($\beta = 0$) (red solid line) is a full-information method using a single gradient query per round. It has a regret bound of $\tilde{O}(T^{2/3})$. Compared to the baseline **Mono** (orange dashed line), which is also a full-information method using a single gradient query per round, our **GMFW**($\beta = 0$)

performed significantly better across different dimensions and horizons, with as little as one seventh of the regret of **Mono**($\beta = 0$) in some experiments.

Comparing our **GMFW**($\beta = 0$) with **Meta**($\beta = 1$) (solid red and dashed red respectively), both of which have $\tilde{O}(T^{2/3})$ regret bounds, we see **Meta**($\beta = 1$) has significantly lower cumulative regret, though the gap between them shrinks in experiments for higher dimensions. As shown in Table 3, however, for a horizon of $T = 100$ with dimension $n = 50$ and $m = 50$ constraints, on average our **GMFW**($\beta = 0$) took less than a second while **Meta**($\beta = 1$) took about one and half minutes.

Lastly, we remark that our semi-bandit **SMFW** (solid blue line), which also uses only a single gradient query evaluated at the action $\mathbf{y}_t$ played, has a regret bound of $\tilde{O}(T^{3/4})$ and empirically has among the highest empirical regrets. Our semi-bandit **SMFW**'s empirical regret is similar to the baseline **Mono** (full information, single gradient query; dashed orange line). Our semi-bandit **SMFW**'s empirical regret is also similar to the baseline **ODC**'s (dotted blue line) regret for low horizons and significantly better for large horizons. **ODC** has the same regret bound, uses full-information feedback, uses more gradient queries, and has much higher computational complexity. As shown in Table 3, for a horizon of $T = 100$ and dimension $n = 50$, for instance, our semi-bandit **SMFW** takes on average about $0.3$ seconds while **ODC** takes almost a minute.

