# OpenReview forum: "Unified Projection-Free Algorithms for Adversarial DR-Submodular Optimization"
_ICLR.cc/2024/Conference — ICLR 2024 poster_

### Official Review · Reviewer_EVLt · 2023-10-28

**Soundness:** 3 good
**Presentation:** 2 fair
**Contribution:** 3 good
**Rating:** 6
**Confidence:** 3

**Summary:**

The paper presents unified projection-free Frank-Wolfe algorithms for online continuous adversarial DR-submodular optimization. It covers various scenarios such as full-information and (semi-)bandit feedback, monotone and non-monotone functions, different constraints, and types of stochastic queries. The algorithms achieve state-of-the-art or improved regret bounds compared to existing methods. The paper also addresses semi-bandit and bandit feedback for adversarial DR-submodular optimization.

**Strengths:**

The paper presents a unified approach for online continuous adversarial DR-submodular optimization, covering a broad spectrum of scenarios. This comprehensive investigation is novel and expands the understanding of the field beyond previous research. The authors introduce technical novelties, such as the combination of meta-actions and random permutations, contributing to the originality of the research.

The paper is well-structured and clearly presents the problem formulation, algorithms, and theoretical analysis. The inclusion of tables and figures enhances the clarity of the presentation.

In summary, the research addresses a significant problem in the field of continuous adversarial DR-submodular optimization with numerous real-world applications. The proposed algorithms demonstrate practical relevance and significance, contributing to the development of efficient optimization techniques for DR-submodular functions.

**Weaknesses:**

While the paper mentions the applications of continuous adversarial DR-submodular optimization, it lacks a thorough empirical evaluation of the proposed algorithms on real-world applications. Including experiments that demonstrate the performance of the algorithms in practical scenarios would strengthen the paper's claims and provide empirical evidence of their effectiveness

I guess that the authors have adopted a text-wrapping layout around the pseudocode due to space constraints. However, the limited space allocated to the pseudocode has resulted in somewhat disorganized format. I would suggest that the authors consider using algorithm2e to reduce the space occupied by the pseudocode, thereby enhancing its readability. Just a friendly suggestion.

**Questions:**

Please refer to weaknesses.

---

> ### Author Response · Authors · 2023-11-20
>
> Thanks a lot for your time and the great comments.
>
> Regarding experiments, we note that the focus of the paper is on the theoretical results.
> We have provided evaluations on non-convex/non-concave quadratic maximization which has also been used in the prior works.
> We note that we consider a unified approach for 16 cases in this paper.
> In 9 of these cases, ours is the first algorithm, and thus there are no baselines to compare.
> Furthermore, one would need to find different real world examples for different settings, since each real-world example often matches the assumption of only a few of the 16 cases.
> While practically important, we chose a case where the assumptions are satisfied to demonstrate the benefits of our approach and where we had relevant baselines to perform the comparison.
> Thus, we leave this comparison to the experts in different domains (e.g., profit maximization in social networks, advertising, experimental design, resource allocation, mean-field inference in probabilistic models, MAP inference in determinantal point processes, etc) where such algorithms could be used in realistic setups.
>
> Regarding the text-wrapping layout, that is indeed due to the space constraint.
> We have followed your advice and have used the algorithm2e package in the revision to enhance readability.

---

### Official Review · Reviewer_5yTK · 2023-10-28

**Soundness:** 3 good
**Presentation:** 2 fair
**Contribution:** 2 fair
**Rating:** 5
**Confidence:** 4

**Summary:**

This paper investigates online adversarial continuous DR-submodular optimization. The authors propose unified projection-free Frank-Wolfe type algorithms under many settings, e.g., full information and (semi-)bandit feedback, monotone and non-monotone functions, different constraints, and types of stochastic queries. Compared with the related works, most of the proposed algorithms achieve state-of-the-art $\alpha$-regret bound.

**Strengths:**

* This paper conducts a detailed investigation on online adversarial continuous DR-submodular optimization and proposes effective algorithms for various settings.
* Compared with existing studies, the proposed algorithms achieves state-of-the-art $\alpha$-regret bound in most settings.

**Weaknesses:**

The presentation of technical contributions and motivation in this paper requires further improvement. See **Questions** below.

**Questions:**

I have the following questions to ask the authors:

* Could you elaborate on the technical challenges of this paper? The proposed algorithms seem to combine existing techniques without introducing novel technical tools.
* This paper proposes two unified projection-free Frank-Wolfe type algorithms for the full-information and bandit settings. Could you provide the reasons why previous algorithms couldn't achieve a unified framework?
* Compared with Zhang et al. (2023) for non-monotone functions, the proposed algorithms achieve only a slight improvement in terms of $\alpha$-regret guarantee. Can you explain the reasons for this, and the differences between two algorithms?
* Could the authors provide a discussion on the technical contributions of this paper regarding online adversarial continuous DR-submodular optimization?

---

> ### Author Response · Authors · 2023-11-16
>
> Thank you for your time in reviewing our paper.
>
> **Q1**: As we  mentioned in the introduction and elaborated on in Sections 3.1-3.2 and Appendix A.2, our main technical novelties include:
> **(i)** a novel combination of the idea of meta-actions and random permutations to obtain a new algorithm in the full-information setting;
> **(ii)** handling stochastic (gradient/value) feedback *without* using variance reduction techniques like momentum, which in turn leads to state of the art regret bounds; and **(iii)** a unified approach that specializes to multiple scenarios considered in this paper.
> (In our submission, we described novelties (ii) and (iii) together; for clarity, we now describe them separately.)
>
> **(i)**
> Except for the full-information algorithm of (Niazadeh et al., 2021) which uses a completely different approach,
> all of the referenced algorithms in prior works designed for the full-information setting with $T^\beta > 1$ gradient samples available for each function $F_t$ used meta-actions, specifically with $K=T^\beta$ online linear optimization (OLO) oracles.
> So the number of oracles $K$ used was set equal to the number of samples $T^\beta$ available as feedback.
> We prove that using $K = T^\beta$ OLO oracles is sub-optimal.
> We do so by designing and analyzing the first random permutations plus meta-action based procedure for this setting.
> Using random permutations, we add a degree of freedom where we can choose the number $K$ of OLO oracles separately from the number of queries $T^\beta$.
> This results in significant improvements in regret bounds.
> For example, in the limit of the number of gradient queries per function $T^\beta$ tending to one, for monotone functions over convex sets containing the origin (Meta-Frank-Wolfe in (Chen et al., 2018a)) and for non-monotone functions over downward closed sets (Meta-MFW in (Zhang et al., 2023)),
> our novel approach using random permutations results in an improvement of $O(T^{1/5})$.
> (Note that this is the improvement due solely from combining random permutations with meta-actions, without considering the effect of removing momentum, which also reduces regret and which we will describe in detail below.
> In the aforementioned settings, when the number of gradient queries per function $T^\beta$ is more than one, our final result improves the state of the art by a factor of $O(T^{1/3})$.)
>
> Intuitively, our approach harmonizes "Meta" algorithms (designed for $T^\beta > 1$ gradient samples per function $F_t$) with "Mono" algorithms (designed for a single, $T^\beta = 1$, gradient sample per function $F_t$), which have never been combined before.
> We strove for our methods to be as simple as possible---the simplicity in combining these ideas is only evident in hindsight and is not evident from the prior works,
> despite the regret bounds for Meta-algorithms as $T^\beta\to 1$ being clearly sub-optimal.
> Even in the most recent work with both types of algorithms,
> (Zhang et al., 2023), separate Meta and Mono algorithms were proposed and they were also  analyzed separately, despite a regret bound gap for $\beta=0$ (see Figure 1 in our paper with $\beta=0$ for a visual depiction of (Zhang et al., 2023)'s Meta-MFW algorithm's regret bound gap compared to that of their Mono-MFW algorithm).
>
> In regards to **(ii)**, one of our main technical novelties is to handle stochastic feedback without relying on variance reduction techniques.
> In particular, not only do we show for all cases that noisy feedback (gradients/values) can be handled without momentum, but more so for a fixed number of queries (fixed $T^\beta$), using momentum leads to higher regret.
>
> Almost all previous Frank-Wolfe type algorithms in online DR-submodular maximization that handle stochastic feedback employ some form of momentum (a common variance reduction method).
> We believe this was largely due to a misinterpretation of the counter-example described in (Hassani et al., 2017) where they show that in the offline setting, directly replacing the exact gradient with an unbiased estimator of the gradient could result in arbitrarily bad outputs of the algorithm.
> This is mentioned in Section 3.2 of (Chen et al., 2018b) as the reason why their version of Meta-Frank-Wolfe may not be used with a stochastic gradient and is later solved in (Chen et al., 2018a) by using momentum which results in considerable loss in performance.
> The version of Meta-Frank-Wolfe described in (Mualem et al., 2023), for non-monotone functions over general convex, works with stochastic gradient oracles without using momentum, which partially inspired this technical novelty of our work.
> However, we note that their work focused only on regret bounds, not query complexity, and if query complexity is ignored it does not matter whether momentum is used-- $O(T^{1/2})$ can be achieved.

---

> ### Author Response · Authors · 2023-11-16
>
> (The fact that their algorithm obtains $O(T^{1-\beta})$ regret for $T^{\beta}$ queries per function follows from the first part of Theorem 4.1 together with the last sentence of Section 4 and the first paragraph of the proof Theorem 3.1 in their paper.)
> While they state their algorithm and proof for an arbitrary number of queries per function, they have no analysis or discussion on query complexity, nor on the possibility of extending that approach to other settings.
> To the best of our knowledge, our paper is the first to realize that removing momentum could result in improved regret bound for a fixed query complexity.
>
> We prove that for online DR-submodular optimization, in all cases, even when the feedback is stochastic, using meta-actions alone (i.e., without momentum) is *sufficient* to obtain sublinear $\alpha$-regret bounds and using momentum decreases the performance.
> We extend this analysis to Mono-Frank-Wolfe, where we improve the state of the art by $O(T^{2/15})$.
> If the idea of random permutations is used as we have described in (i), together with momentum, the regret will likely be bounded from above by $O(T^{4/5 - \beta/5})$.
> By removing momentum, we achieve $O(T^{2/3 - \beta/3})$ regret, i.e. a likely improvement of $O(T^{2(1 - \beta)/15})$ over the result one would obtain by using (i) with momentum.
>
> For **(iii)**, we note that many papers and results add more complexity to the field of research, but our result tries to move in the opposite direction. Algorithm 1 (for full-information feedback) generalizes, simplifies and improves 8 previous algorithms, while Algorithm 2 (for semi-bandit feedback) does this for 3 previous algorithms.
> We developed a unified analysis which is more modular and simpler than analyses in prior works.
> This modularity allows us to *(a)* obtain better insight for how elements of the algorithm design and problem setting interact in bringing about the $\alpha$ approximation ratio separately from the $\alpha$-regret bounds; and as consequence, results in *(b)* proof for regret guarantees for 16 algorithms in shorter space than other papers proved regret guarantees for just one or two algorithms.
>
> Regarding *(iii).(a)* on how the modularity of the analysis yields clearer insight,
> in Appendix E we show an essential lemma (Lemma 8) that makes explicit why the update rules and the step sizes should be the way they are.
> Lemma 8 only considers a single DR-submodular function and captures how certain update rules result in certain approximation coefficients.
> In fact, besides the two algorithms in this paper, Lemma 8 may be directly used in the proof of at least 18 algorithms in the literature as the part of the proof which derives the approximation coefficients: (Chen et al., 2018b) (Chen et al., 2018a) (Zhang et al., 2019) (Niazadeh et al., 2021) (Thang et al., 2021) (Zhang et al., 2023) (Mualem et al., 2023) (Bian et al., 2017b) (Mokhtari et al,. 2020) (Bian et al., 2017a) (Du et al., 2022) (Pedramfar et al., 2023)
>
> Regarding *(iii).(b)*, as an example of how much the modularity of our analysis leads to brevity, in the paper (Zhang et al., 2019), where the idea of random permutations is first used for DR-submodular maximization with full-information single sample gradient $T^\beta=1$ feedback, the proof of their result for the full-information setting
> is 8.5 pages (Appendix B in their paper).
> That proof only covers the  (simpler) case of monotone functions over convex sets containing the origin, with  a single evaluation of gradient oracle.
> They use another 9 pages to prove their result for monotone functions over downward closed convex sets in the bandit setting.
> (We are only considering the proofs that involve the ideas of meta-actions and random permutations, not general statements about convex sets or DR-submodular functions.)
> In contrast, the only part of our proofs that deals explicitly with the online setting are in Appendix F and G, a total of 4 pages, which, together with Lemma 9, prove the regret bounds for 16 problem variations listed in Table 1.
>
> **Q2**: We cannot speak for authors of other papers on why they did not propose a unified framework.
> Instead, we partly address a related question -- in what way is this work unified and what more work would some authors have needed to do in order to unify their separately described and analyzed algorithms.
> In general, that would depend on how one defined the notion of "unification".
> There are several different forms of unification appearing in our work:
>
> **(a)** Unification of value/gradient oracle settings:
> Algorithm 1 is the first algorithm for adversarial DR-submodular maximization with full information feedback given a value oracle.
> Except for (Chen et al., 2018b) which uses a projection-based method, Algorithm 2 is the first algorithm for adversarial DR-submodular maximization with semi-bandit feedback given a gradient oracle.
> This is arguably the simplest form of unification present in our work.

---

> > ### Author Response · Authors · 2023-11-16
> >
> > **(b)** Unification of adversarial and stochastic bandit feedback settings:
> > In this work, we consider the setting of *noisy* adversarial bandit feedback which is a generalization of adversarial bandit feedback and stochastic bandit feedback.
> > Note that, in previous works, momentum was used to move from a deterministic gradient oracle to a stochastic one.
> > Here we need to move from a deterministic value oracle to a stochastic one.
> > One of our main technical novelties is to show that momentum is not needed to handle a stochastic oracle. See the answer to Q1-(ii) for more details.
> >
> > **(c)** Unification of algorithms for monotone/non-monotone functions and different constraint sets:
> > This type of unification is not necessarily too difficult if we are not concerned with the performance of the algorithm or having separate proofs for each case.
> > However, without using the ideas described in the responses to Q1-(i) and Q1-(ii), the guarantees will be worse than ours even if we use such a unification in previous works.
> > Please see the following paragraph for more details.
> >
> > **(d)** Unification of Meta-Frank-Wolfe and Mono-Frank-Wolfe:
> > See the answer to Q1-(i).
> >
> > **(e)** Unification of the proofs:
> > See the answer to Q1-(iii).
> >
> > More details on **(c)**:
> >
> > Using the update rules described in (Bian et al., 2017b) (Bian et al., 2017a) (Du et al., 2022) and (Pedramfar et al., 2023), one may generalize previous results to include cases with monotone/non-monotone functions given different constraint sets.
> > First we consider the full-information feedback setting.
> >
> > - Using the aforementioned update rules, the result of (Chen et al., 2018a) may be generalized to a unified approach in the sense of (c).
> > However, the performance would be significantly worse than our results, by a factor of $O(T^{1/3})$.
> > The same could be said of the full-information result of (Niazadeh et al., 2021).
> >
> > - Similar to above, one may generalize the results of (Mualem et al., 2023) to other cases.
> > This would lead to a unified approach with a performance that would be $O(T^{1/3 - 2\beta/3})$ worse than ours.
> >
> > Note that in these algorithms, if we are only allowed to query each functions $O(1)$ times, we would obtain a linear regret.
> >
> > - The Mono-Frank-Wolfe of (Zhang et al., 2019) is not a general algorithm as it only considers the case where we are allowed to query each function once.
> > Merging this algorithm with Meta-Frank-Wolfe is one of our main technical novelties (see our response to Q1-(i)) which would (likely) result in a unified approach with a performance that is $O(T^{2(1 - \beta)/15})$ lower than Algorithm 1.
> > We finally reach our result if we merge Mono-Frank-Wolfe with Meta-Frank-Wolfe and also remove momentum (see our response to Q1-(ii)).
> >
> > *We note that these algorithms are for the full-information setting with gradient oracles.
> > Our Algorithm 1 takes unification a step further by considering all full-information settings, including the setting where we only have access to stochastic value oracles.*
> >
> > Next we consider the (semi-)bandit feedback setting.
> >
> > - Similar to above, the Bandit-Frank-Wolfe algorithm of (Zhang et al., 2019) may be generalized to other case.
> > In fact, one of the main results of (Zhang et al., 2023) was extending this to the case of non-monotone functions over downward-closed convex sets.
> > This would lead to a unified approach with a performance that is $O(T^{1/18})$ lower than ours.
> >
> > - On the other hand, the bandit feedback result of (Niazadeh et al., 2021) can likely be generalized with the same regret bound as ours.
> >
> > *However, we note that all of these algorithms are for adversarial bandit feedback setting and do not apply to stochastic bandit feedback setting. (see (b) above)*
> >
> >
> > **Q3**: We would like to gently push back on the statement "the proposed algorithms achieve only a slight improvement".
> > Note that we are aiming for sublinear regret, i.e., $o(T)$, and we do not expect the regret to be lower than $O(T^{1/2})$. Hence the whole range for improvement is from $O(T)$ to $O(T^{1/2})$.
> > (Zhang et al., 2023) proposed 3 algorithms: Meta-MFW, Mono-MFW, and Bandit-MFW.
> > Meta-MFW obtains a regret of $O(T^{1-\beta/3})$ with $T^{\beta}$ gradient queries. We obtain $O(T^{2/3 - \beta/3})$ which is an improvement of order $O(T^{1/3})$.
> > Mono-MFW obtains a regret of $O(T^{4/5})$ given a single gradient query. We obtain $O(T^{2/3})$ in the same setting which is an improvement of order $O(T^{2/15})$.
> > Finally, Bandit-MFW obtains a regret of $O(T^{8/9})$ given bandit feedback. We obtain a better bound of $O(T^{5/6})$ in a more general setting, i.e., noisy bandit feedback, which is an improvement of order $O(T^{1/18})$.
> > Please see our responses to Q1-(i) and Q1-(ii) regarding the differences in algorithm design.
> >
> > **Q4**: Please refer to our response to your Q1 above for novelty, Section 3.1 and 3.2 for more details (specifically the ideas of meta-actions and random permutations) and Appendix A.2 for a comprehensive review of previous works.

---

> > > ### Comment · Reviewer_5yTK · 2023-11-19
> > > **Thanks for the detailed responses.**
> > >
> > > The thorough discussions have resolved the issues I raised in Q2 and Q3. However, I am still confused about the technical challenges in this paper. First, I believe that a "novel combination" in online learning cannot be considered a technical contribution. Second, the paper primarily uses existing techniques, i.e., random permutations to achieve a series of improved bounds, as the authors have mentioned in rebuttal. Finally, technical novelties (ii) and (iii) also do not introduce any novel technical tool but simply combine existing techniques and provide corresponding analysis. In this review round, I will maintain my score, reflecting my uncertainty about the technical contributions of this paper.

---

> > > > ### Author Response · Authors · 2023-11-20
> > > >
> > > > We note that technical novelty is subjective, and what one considers innovative or novel can vary based on individual perspectives within the research community. Some researchers may emphasize the introduction of entirely new tools, while others might appreciate creative combinations of existing ideas or the application of known methods in a new context. In that sense, many open problems have been resolved by creative combinations of existing ideas, and thus is perceived to be novel by the research community.
> > > >
> > > > Consider for example the Chernoff bound, which was first described in a 1952 paper, decades after Markov and Chebyshev.
> > > > It provides a new insight which allows for sharper concentration bounds compared to Markov's inequality.
> > > > We consider the idea of Chernoff bound to be a technical novelty, even though the result is an almost immediate corollary of Markov's inequality.
> > > > Our approach of unified algorithm and analysis extends multiple ideas beyond the settings they were originally designed for, and gives new insights to the area. These insights allowed us to improve the state of the art in almost all cases.
> > > >
> > > > Oftentimes the difference between "a novel combination of existing ideas" and "a new idea" is in having a new name.
> > > > Many times in machine learning, we have seen that old ideas have been used with a new name and considered novel.
> > > > However, we have tried to give credit to previous works as much as possible. In fact, if we are being precise, 9 out of the 16 previous algorithms mentioned in Table 1 either require stronger assumptions to work (See Appendix C on the boundedness of stochastic oracles, note that we have updated our paper) or have proved less than we have attributed to them by putting them in that table (See Appendix C for details).
> > > >
> > > > In regards to the novelty of (ii), we note that realizing that a tool is not needed is by itself a novelty.
> > > > For example, consider the idea of Generative Pre-Trained (GPT) models.
> > > > GPT models were introduced in (Improving Language Understanding by Generative Pre-Training, Alec Radford et al., 2018) using only the decoder half of the Transformer model (Attention Is All You Need, Vaswani et al., 2017). They demonstrated that the attention mechanisms in the decoder layers enable the model to capture dependencies and context over long ranges, contributing to its effectiveness in various language-related applications.
> > > >
> > > > We also note that the unification of multiple scenarios often involves recognizing connections and similarities between seemingly unrelated ideas.
> > > > Unification is considered a novelty because it reflects the discovery of hidden connections, the creation of more general frameworks, and the ability to simplify and illuminate complex landscapes.
> > > >
> > > > Finally, we note that (Chen et al., 2018b) combined the idea of meta-actions (Streeter et al., 2008) and the update rule of (Bian et al., 2017b) to obtain Meta-Frank-Wolfe.
> > > > (Chen et al., 2018a) combined the idea of (Chen et al., 2018b) and the momentum idea described in (Mokhtari et al., 2020).
> > > > (Mualem et al., 2023) combined the update rule of (Du et al., 2022) and the Meta-Frank-Wolfe idea of (Chen et al., 2018b) to obtain their algorithms.
> > > > (Zhang et al., 2023) combined the idea of (Chen et al., 2018b) and (Zhang et al., 2019) and a slightly modified version of the update rule in (Bian et al., 2017a) to obtain their results.
> > > > We respectfully mention that based on the reviewer's comment, it feels that the reviewer is saying none of these results have any technical novelty since most of the tools were in existing literature, and should not have been accepted to top-tier venues.
> > > > We believe that they are novel because they describe approaches to use known tools to obtain new results, and those combinations of ideas have not been thought of before.

---

### Official Review · Reviewer_ZRDt · 2023-11-01

**Soundness:** 3 good
**Presentation:** 4 excellent
**Contribution:** 4 excellent
**Rating:** 10
**Confidence:** 3

**Summary:**

The paper proposes an expedient and highly generalized projection-free Frank-Wolfe type algorithms for adversarial and continuous DR-submodular maximization problems. The settings they consider can be a full information or a (semi-) bandit feedback, with monotone or non-monotone functions, having different constraints, and types of stochastic queries. Notably, for every problem they consider in the non-monotone setting, they either provide the first algorithm with proven sub-linear $\alpha$-regret bounds or they improve the already existing $\alpha$-regret bounds. They also report obtaining the state-of-the-art sub-linear $\alpha$-regret bounds among projection-free algorithms in the majority of the monotone settings they consider. In the remaining monotone setting, they match their result to the existing one.

**Strengths:**

The paper's efforts to present unified projection-free algorithms is convenient and worthy of exploration. The main technical contributions of the paper, a.k.a. combining the ideas of meta-actions and random permutations and providing a refined analysis that does not rely on variance reduction techniques, are original. The ideas, contributions, and techniques are expressed clearly. I especially enjoyed the level of detail in Table 1 and Appendices A.2 and C.

**Weaknesses:**

I have not noticed particular weaknesses so far.

**Questions:**

I do not have any clarifying questions on my mind at the moment.

---

> ### Author Response · Authors · 2023-11-16
>
> We genuinely appreciate your time and effort in reviewing our work. Thank you very much for your valuable feedback and support.

---

### Meta-Review · Area_Chair_429N · 2023-12-12

**Metareview:**

The paper studies the problem of maximizing a continuous submodular function in the online setting. The paper studies the problem in a wide range of settings, including both monotone and non-monotone objective functions, down-monotone and general convex domains, and several feedback models and types of feedback. Using a unified algorithmic approach, the paper gives sublunar regret guarantees in all of these settings, improving upon the state of the art guarantees for projection-free algorithms in most of the settings considered.

The reviewers were divided regarding the strength and novelty of the proposed approach. Two of the reviewers appreciated the strength and generality of the proposed algorithm and the improvements in the regret obtained in a wide range of settings. One of the reviewers remained concerned about the novelty of the techniques and the strength of the improvement.

As noted in the author response and also summarized in Table 1, the paper improves upon the prior regret guarantees in most of the settings considered, using a simpler approach that does not employ momentum or variance reduction. The algorithmic approach and analysis techniques also have novel elements compared to prior work, as summarized in the author response and the paper. Overall, this work makes a meaningful contribution to this line of work.

**Justification For Why Not Higher Score:**

Although the paper's contributions are solid and relevant, they may not rise to the level of a spotlight presentation.

**Justification For Why Not Lower Score:**

The paper makes a valuable contribution to this line of work.

---

### Decision · Program_Chairs · 2024-01-16

Accept (poster)